# Exploring the Biomedical Applications of Biosynthesized Silver Nanoparticles Using *Perilla frutescens* Flavonoid Extract: Antibacterial, Antioxidant, and Cell Toxicity Properties against Colon Cancer Cells

**DOI:** 10.3390/molecules28176431

**Published:** 2023-09-04

**Authors:** Tianyu Hou, Yurong Guo, Wanyu Han, Yang Zhou, Vasudeva Reddy Netala, Huizhen Li, He Li, Zhijun Zhang

**Affiliations:** School of Chemistry and Chemical Engineering, North University of China, Taiyuan 030051, China; Gyr_201605@163.com (Y.G.); m13152990286_2@163.com (W.H.); 1356997773@163.com (Y.Z.); vasunuc1922@gmail.com (V.R.N.); hzli@nuc.edu.cn (H.L.); heli_science@nuc.edu.cn (H.L.)

**Keywords:** *Perilla frutescens*, AgNPs, TEM, antibacterial, anticancer, COLO205

## Abstract

The present study reports the biomimetic synthesis of silver nanoparticles (AgNPs) using a simple, cost effective and eco-friendly method. In this method, the flavonoid extract of *Perilla frutescens* (PFFE) was used as a bioreduction agent for the reduction of metallic silver into nanosilver, called *P. frutescens* flavonoid extract silver nanoparticles (PFFE-AgNPs). The Ultraviolet–Visible (UV-Vis) spectrum showed a characteristic absorption peak at 440 nm that confirmed the synthesis of PFFE-AgNPs. A Fourier transform infrared spectroscopic (FTIR) analysis of the PFFE-AgNPs revealed that flavonoids are involved in the bioreduction and capping processes. X-ray diffraction (XRD) and selected area electron diffraction (SAED) patterns confirmed the face-centered cubic (FCC) crystal structure of PFFE-AgNPs. A transmission electron microscopic (TEM) analysis indicated that the synthesized PFFE-AgNPs are 20 to 70 nm in size with spherical morphology and without any aggregation. Dynamic light scattering (DLS) studies showed that the average hydrodynamic size was 44 nm. A polydispersity index (PDI) of 0.321 denotes the monodispersed nature of PFFE-AgNPs. Further, a highly negative surface charge or zeta potential value (−30 mV) indicates the repulsion, non-aggregation, and stability of PFFE-AgNPs. PFFE-AgNPs showed cytotoxic effects against cancer cell lines, including human colon carcinoma (COLO205) and mouse melanoma (B16F10), with IC_50_ concentrations of 59.57 and 69.33 μg/mL, respectively. PFFE-AgNPs showed a significant inhibition of both Gram-positive (*Listeria monocytogens* and *Enterococcus faecalis*) and Gram-negative (*Salmonella typhi* and *Acinetobacter baumannii*) bacteria pathogens. PFFE-AgNPs exhibited in vitro antioxidant activity by quenching 1,1-diphenyl-2-picrylhydrazyl (DPPH) and hydrogen peroxide (H_2_O_2_) free radicals with IC_50_ values of 72.81 and 92.48 µg/mL, respectively. In this study, we also explained the plausible mechanisms of the biosynthesis, anticancer, and antibacterial effects of PFFE-AgNPs. Overall, these findings suggest that PFFE-AgNPs have potential as a multi-functional nanomaterial for biomedical applications, particularly in cancer therapy and infection control. However, it is important to note that further research is needed to determine the safety and efficacy of these nanoparticles in vivo, as well as to explore their potential in other areas of medicine.

## 1. Introduction

Nanobiotechnology, an interdisciplinary field that merges biology, chemistry, and physics, has recently emerged as a promising discipline with a wide range of applications [1]. One of the key areas of nanobiotechnology is the synthesis and characterization of nanoparticles, which are particles with sizes below 100 nanometers. Two major approaches for synthesizing nanoparticles are the top-down and bottom-up approaches, with the latter becoming increasingly popular due to its simplicity, cost-effectiveness, and robustness [2]. Metal nanoparticles possess unique properties such as optical, magnetic, catalytic, mechanical, electronic, and thermal properties, which stem from their surface energy, spatial confinement, and high surface-area-to-volume ratio [3,4]. The localized surface plasmon resonance (SPR) and surface-enhanced Raman scattering (SERS) of metal nanoparticles make them highly attractive for use in next-generation electronic and biochemical sensors [5,6]. In this context, the development of metal nanoparticles and their integration into biological systems is opening new avenues for research in nanobiotechnology. By exploring the properties of metal nanoparticles, researchers are developing innovative applications for them in various fields, including drug delivery, imaging, sensing, and therapy.

AgNPs have been widely employed in different fields, including as optical receptors [4,5,6], intercalation materials for electrical batteries [7], catalysts in chemical and biochemical reactions [8], sensors and biosensors [9,10], bio-labeling materials [11], signal enhancers in SERS-based enzymes for immunoassay [12], and antimicrobial agents [13]. Due to recent advancements in nanotechnology, AgNPs have been successfully employed for cancer therapy, tissue engineering, drug delivery, inflammation, tuberculosis, diabetes, cardiovascular diseases, autoimmune disorders (Rheumatoid arthritis), and neurodegenerative disorders (Parkinson’s and Alzheimer’s) [14,15,16,17,18,19,20]. 

The unique physicochemical properties of metal nanoparticles will be determined, including their shape, size, crystallinity, dispersity, and surface charge. Hence, the synthesis of monodispersed AgNPs with ultra sizes and different shapes is very essential. Different chemical and physical approaches including UV-irradiation [21], gamma irradiation [22], ultrasound irradiation [23], thermal decomposition [24], laser ablation [25], aerosol [26], lithographic [27], electrochemical assisted [28], sonochemical synthesis [29], polyol [30], polyaniline [31], and chemical reduction [32] approaches have been widely employed to produce AgNPs. However, these approaches are not eco-friendly due to the application of irradiation, hazardous substances and toxic chemicals. Further, they are time-consuming, laborious and expensive. Further, the toxic chemicals on the surface (capping agents) of the AgNPs limit their applications in biomedical and diagnostic fields [33]. Hence, the production of eco-friendly, non-toxic, clean, biocompatible, and biofunctionalized AgNPs using biological agents deserves merit. 

In recent years, the field of nanotechnology has witnessed a remarkable convergence with the principles of biomimicry, giving rise to a revolutionary approach known as the biomimetic synthesis of AgNPs [34]. This innovative synthesis method draws inspiration from the intricate processes found in nature, harnessing the power of biomolecules and their interactions to fabricate AgNPs with unprecedented precision and control. These nanoparticles, often at the nanoscale level, exhibit unique physical, chemical, and biological properties that hold immense promise for a wide range of biomedical applications [35]. The biomimetic synthesis of AgNPs involves the emulation of biological systems and their underlying mechanisms, such as enzymatic reactions or self-assembly processes, to guide the formation and stabilization of these nanoparticles [36]. This novel approach offers distinct advantages over conventional methods, enabling the production of nanoparticles with well-defined sizes, shapes, and surface characteristics. Moreover, the integration of biological molecules into the synthesis process enhances the biocompatibility and functionality of the resulting AgNPs, making them highly suitable for various biomedical applications [37]. Biosynthesized AgNPs offer several distinct advantages over their chemically synthesized counterparts, making them particularly well-suited for a wide range of biomedical applications [34,35]. These advantages stem from the unique properties and characteristics that result from the biomimetic synthesis process. Here are some key points of comparison between biosynthesized AgNPs and their chemical counterparts in the context of biomedical applications [38]. Biosynthesized AgNPs often involve the use of natural biomolecules, such as proteins, enzymes, or plant extracts, as reducing and stabilizing agents. These biomolecules are typically biocompatible and non-toxic, which translates into reduced potential for adverse effects when used in biological systems [39]. In contrast, chemically synthesized AgNPs may involve the use of harsh reducing agents or stabilizers that can introduce toxicity concerns [34,35,36]. Biomimetic synthesis methods frequently enable precise control over the size, shape, and morphology of AgNPs. This level of control is challenging to achieve with chemical synthesis methods, which can result in a broader distribution of particle sizes and shapes [40]. The ability to produce nanoparticles with specific attributes is crucial for applications where size and shape play a significant role, such as targeted drug delivery or imaging agents [41].

Biomimetic synthesis often allows for the facile functionalization of the nanoparticle surfaces with biomolecules, antibodies, or other ligands [42]. This functionalization enhances the nanoparticles’ ability to interact selectively with specific cells, tissues, or biomolecules, facilitating targeted drug delivery, imaging, and sensing [43]. Chemical synthesis methods may require additional steps and modifications to achieve comparable functionalization. Biosynthesized AgNPs tend to exhibit enhanced stability due to the presence of biomolecules that contribute to their robustness in various environmental conditions [44]. This stability is especially crucial for biomedical applications where nanoparticles need to maintain their integrity during storage, transport, and interactions with biological systems [45]. The biomimetic synthesis of AgNPs often employs renewable resources and green chemistry principles. This aligns with the growing emphasis on sustainable and environmentally friendly practices. In contrast, chemical synthesis methods may involve the use of hazardous chemicals and energy-intensive processes [33,34,35]. Biomimetic synthesis methods can lead to nanoparticles with improved dispersibility and reduced agglomeration, which is essential for maintaining consistent behavior and interactions within biological systems. Agglomerated nanoparticles may not distribute evenly or exhibit the desired properties, limiting their effectiveness in biomedical applications [46]. The use of biomolecules in biosynthetic processes opens the door to incorporating multifunctional features into the nanoparticles. For instance, enzymes or peptides can confer catalytic or targeting properties to the nanoparticles, respectively. Such multifunctionality is challenging to achieve using chemical synthesis alone [47].

Active ingredients and plant extracts are often natural and biocompatible, making them ideal candidates for use in biomedical applications [48]. When utilized in nanoparticle synthesis, these natural compounds can improve the biocompatibility of the nanoparticles, reducing the risk of toxicity and adverse reactions in medical or biological settings [49]. Active ingredients and plant extracts contain various bioactive compounds that can endow nanoparticles with unique functionalities [20,48,49,50]. These functionalities can include antibacterial, antifungal, antioxidant, anti-inflammatory, and other therapeutic properties, depending on the specific plant extracts used. Such functionalized nanoparticles have promising applications in medicine, agriculture, and environmental remediation [51]. Combining different active ingredients or plant extracts in nanoparticle synthesis can lead to synergistic effects where the resulting nanoparticles exhibit enhanced or novel properties compared to the individual components [52]. Such synergies may improve the overall performance and efficiency of nanoparticles in various applications [53]. The biodegradability of active ingredients and plant extracts is a desirable property when considering the potential biomedical applications of nanoparticles [54]. Biodegradable nanoparticles can be designed to release their payload in a controlled manner, which is particularly advantageous for drug delivery systems. By incorporating active ingredients and plant extracts, researchers can opt for a more sustainable and eco-friendly approach to nanoparticle synthesis [55]. Further, plant extracts are abundantly available, making them cost-effective options for nanoparticle synthesis. By using readily accessible natural sources, researchers can reduce production costs and increase the feasibility of large-scale nanoparticle manufacturing [56].

In the context of AgNPs, recent reports have highlighted the successful use of various plant extracts, such as ginger [50], curcumin [57], garlic extract [58], *Allium cepa* peel extract [59], Aloe vera [60], green tea [61], seed extracts *Brassica oleraceae*, *B. campestris* and *B. rapa* [62], and *Teucrium polium* leaf extract [63] as potent reducing agents for the synthesis of AgNPs. These natural extracts contain bioactive compounds, such as polyphenols, flavonoids, and terpenoids, which play a crucial role in the reduction and stabilization of nanoparticles.

Flavonoids are a class of polyphenolic compounds that are widely distributed in the plant kingdom and have a wide range of biological activities, including antioxidant, anti-inflammatory, and antibacterial properties [64]. Flavonoids are used in the synthesis of AgNPs because they have unique chemical properties that make them ideal for reducing silver ions into nanoparticles. Flavonoids are known to have a high affinity for metal ions, and they can act as reducing agents in the presence of metal ions to form nanoparticles [65].

During the biosynthesis of AgNPs, flavonoids are present in the plant extract and act as reducing agents that help in the conversion of silver ions to AgNPs. Flavonoids contain multiple hydroxyl groups that are capable of reducing metal ions to nanoparticles by donating electrons. The hydroxyl groups present in the flavonoids interact with the silver ions, leading to the reduction of the silver ions and the formation of AgNPs [65,66]. Furthermore, flavonoids also act as stabilizing agents for AgNPs, preventing their agglomeration and ensuring their stability in the solution [65,66,67]. The hydroxyl groups present in the flavonoids interact with the surface of AgNPs, forming a layer that stabilizes the nanoparticles and prevents them from aggregating. This makes flavonoids a useful tool in the biosynthesis of AgNPs.

*Perilla frutescens* is a plant species in the mint family (Lamiaceae) that is native to Asia. It is also known as Chinese basil, wild basil, or shiso in Japan. The plant is cultivated for its edible leaves, which are used in various culinary dishes, especially in Korean, Japanese, and Chinese cuisine [68,69]. The leaves are also used in traditional medicine for their medicinal properties. *P. frutescens* contains various bioactive compounds, including flavonoids, phenolic acids, terpenoids, and essential oils, that contribute to its therapeutic properties [70]. Among these, flavonoids are the most studied compounds and have been shown to have various biological activities, including antioxidant, anti-inflammatory, antimicrobial, antiallergic, and anticancer activities [71]. These findings suggest that *P. frutescens* flavonoids may have potential therapeutic applications in the prevention and treatment of various diseases, including cancer, cardiovascular diseases, diabetes, and neurodegenerative disorders. Overall, *P. frutescens* and its flavonoids are promising candidates for the development of novel therapeutics or functional foods for promoting human health and preventing chronic diseases.

The present study involving *P. frutescens* flavonoid extract in the synthesis of AgNPs aims to explore the green synthesis approach, which involves using natural sources instead of conventional chemical methods. Green synthesis offers several advantages, including eco-friendliness, cost-effectiveness, and the potential for producing nanoparticles with unique properties. One of the significant challenges in this area of research is the optimization of the synthesis conditions. The extraction of flavonoids from *P. frutescens* and subsequent reduction of silver ions to form nanoparticles require the careful control of various parameters such as temperature, concentration, and reaction time. Finding the optimal conditions to obtain well-defined nanoparticles with desirable properties can be a complex and time-consuming process.

The primary objective of this investigation is to explore the biomimetic synthesis of AgNPs utilizing the natural flavonoid extract derived from *P. frutescens*, denoted as PFFE. Through this innovative approach, we aim to produce finely tuned PFFE-AgNPs and subject them to thorough characterization using advanced spectroscopic techniques. Furthermore, our study aims to delve into the multifaceted potential of PFFE-AgNPs in diverse biomedical applications. The evaluation will encompass an in-depth analysis of their antibacterial efficacy, aiming to shed light on their ability to combat microbial infections. Additionally, we will investigate their antioxidant properties, seeking insights into their potential to counteract oxidative stress-related damage. Moreover, our investigation extends to exploring the anticancer activities of PFFE-AgNPs, aiming to uncover any potential inhibitory effects on cancer cells. By comprehensively assessing these key aspects, we aspire to contribute to the understanding of PFFE-AgNPs as versatile agents with potential applications in healthcare and biomedicine. This study stands to offer valuable insights into the promising realm of nanotechnology-based therapeutics and their multifunctional roles in addressing critical health challenges.

## 2. Results and Discussion

### 2.1. Biomimetic Synthesis of PFFE-AgNPs

In the present study, we prepared the flavonoid extract using the leaves of *P. frutescens*, referred to as *P. frutescens* flavonoid extract (PFFE). The total flavonoid content (TFC) of the PFFE was determined using the AlCl_3_ colorimetric method. The TFC of the PFFE was found to be 46.8 µg/mL of QEs. The PFFE was employed for the biomimetic synthesis of PFFE-AgNPs. After incubation, the reaction mixture, containing PFFE and 2 mM of AgNO_3_, turned dark brown from a light-yellow color. This color change indicates the formation of PFFE-AgNPs (*P. frutescens*-flavonoid-extract-mediated AgNPs). The color change is due to localized surface plasmon resonance (LSPR). The synthesis of PFFE-AgNPs was further confirmed by a UV-Vis analysis of the colloidal solution.

### 2.2. UV-Vis Analysis of PFFE-AgNPs

The UV-Vis spectrum was recorded between 200 and 800 nm to confirm the biomimetic synthesis of PFFE-AgNPs. The UV-Vis spectrum showed a peak between 350 and 450 nm, with highest absorbance at 440 nm (Figure 1). This is the characteristic absorbance peak of PFFE-AgNPs due to localized surface plasmon excitations by metallic silver in the visible range [1,2,3].

The surface plasmon resonance is a phenomenon that occurs when light interacts with the surface electrons of the AgNPs, causing a collective oscillation of the electrons and resulting in a characteristic absorbance peak in the visible or near-infrared region [1,2,3]. Similar findings were reported for the synthesis of AgNPs using flavonoids from *ocimum sanctum* [66] and *Reinwardtia indica* [67], where the surface plasmon resonance peak is due to the conversion of silver ions into AgNPs. The broad peak in this study is responsible for the spherical shape of PFFE-AgNPs, which is further confirmed by a TEM analysis. In this study the components of PFFE act as reducing agents for the conversion of silver ions into AgNPs in an aqueous medium. Furthermore, these components stabilize the synthesized PFFE-AgNPs by capping them. The components involved in bioreduction and capping were studied using a FTIR functional group analysis.

### 2.3. FTIR Analysis of PFFE-AgNPs

Fourier transform infrared (FTIR) spectroscopy is a useful technique for characterizing the functional groups and chemical bonds present on the surface of AgNPs [72]. FTIR analysis involves directing infrared radiation through a sample and measuring the absorption and transmission of the radiation at different wavelengths. The resulting spectrum can provide information on the functional groups and chemical bonds present in the sample. In the present study, the FTIR spectrum revealed peaks at 895, 1241, 1810, 2968, and 3381 cm^−1^ (Figure 2). The peaks at 1241 and 895 cm^−1^ correspond to O-H and C-O stretching, respectively. The peaks at 1810, 2968, and 3381 cm^−1^ correspond to the C=O, C≡C, and O-H stretching vibrations of aromatic compounds, respectively, which participated in the biosynthesis of PFFE-AgNPs [66,67]. Thus, the FTIR spectrum clearly revealed that the flavonoids are involved in the bioreduction of silver ions into PFFE-AgNPs. FTIR analysis is a valuable technique for characterizing the surface chemistry of AgNPs and can provide insight into the stabilization mechanisms and potential applications of the particles in various fields.

### 2.4. TEM, SAED and XRD Analysis of PFFE-AgNPs

TEM studies were carried out to reveal the size, shape, and SAED pattern of PFFE-AgNPs [73]. The TEM micrograph (Figure 3a) revealed that the biosynthesized PFFE-AgNPs are 20–70 nm in size with a spherical shape. The TEM micrograph also showed that PFFE-AgNPs are monodispersed with no aggregation. The non-aggregation of PFFE-AgNPs might be due to capping. Further, the size distribution, dispersion, and aggregation of AgNPs was studied using a DLS technique. The SAED pattern of PFFE-AgNPs (Figure 3b) showed the Debye–Scherrer diffraction rings responsible for the crystalline nature of PFFE-AgNPs. Further, the crystalline nature was confirmed by an XRD analysis. The TEM results are in line with previous reports for AgNPs synthesized using different flavonoid extracts. The spherical-shaped AgNPs with a size of 15–30 nm were synthesized using *Ocimum sanctum* [66]. Spherical-shaped AgNPs with a size of 3–15 nm were also observed using the flavonoid extract of *Reinwardtia indica* [67].

The XRD analysis was conducted to reveal the crystal structure of the PFFE-AgNPs [74]. The XRD pattern (Figure 4) showed four diffraction peaks at 38.2°, 44.7°, 64.1°, and 77.3°, which are responsible for the planes (1 1 1), (2 0 0), (2 2 0), and (3 1 1), respectively. These Bragg peaks clearly revealed the FCC (face-centered cubic) lattice of nanosilver (JCPDS file No. 87-0597). 

The crystal nature of the AgNPs is an important characteristic feature for their effective permeation through target cells in anticancer therapy. Flavonoid-mediated AgNPs with an FCC crystal lattice were observed using *Ocimum sanctum* [66] and *Reinwardtia indica* [67]. Overall, the TEM, SAED, and XRD analyses are complementary techniques that can provide valuable information on the size, shape, and crystal structure properties of AgNPs. Together, these techniques provide insight into potential applications of AgNPs in various fields.

### 2.5. DLS Analysis of PFFE-AgNPs

The average hydrodynamic size of the PFFE-AgNPs can be determined through a DLS analysis by measuring the time-dependent fluctuations in the intensity of scattered light from the particles [75]. This measurement is based on the Brownian motion of particles in a solution, and the hydrodynamic size is calculated based on the speed of the particles’ motion and their interaction with the surrounding solvent molecules [76]. DLS studies revealed that particles are distributed between 20 and 70 nm in a water dispersion medium (Figure 5). The average hydrodynamic radius of the PFFE-AgNPs was found to be 44.0 ± 14.4 nm (Table 1). This hydrodynamic radius suggests that the AgNPs are relatively small, which is consistent with their unique physical and chemical properties that make them useful in a wide range of applications. The size distribution of the particles, as indicated by the relatively large standard deviation of 14.4 nm, suggests that there may be some variation in the size of the particles within the sample. The polydispersity index (PI) of the PFFE-AgNPs was found to be 0.321. This PI value indicated that the PFFE-AgNPs are monodispersed with a uniform size distribution. The particle size distribution showed that 30% of the PFFE-AgNPs were below 40 nm in size, and 70% of the AgNPs produced were above 40 nm in size.

In addition, DLS analysis can also measure the zeta potential, which is a measure of the electrostatic charge on the surface of the particles [77] (Table 2). The DLS studies revealed that the Zeta potential of PFFE-AgNPs was −30.0 mV (Figure 6). A highly negative zeta potential indicates that the particles are highly negatively charged, which can result in improved stability and reduced aggregation of the particles in the solution [78]. This can be particularly important for the biomedical and environmental applications of AgNPs, in which stability and controlled dispersion are important factors. Overall, DLS analysis is a valuable technique for characterizing the physical properties of AgNPs and understanding their behavior in a solution.

The average hydrodynamic radius, along with other characterization techniques, can be used to understand the behavior of the PFFE-AgNPs in a solution, including their stability, surface charge, and interactions with other molecules or particles [79]. This information is important for the development of applications such as antimicrobial agents, drug delivery systems, and environmental remediation.

### 2.6. Antibacterial Activity of the PFFE-AgNPs

In the present investigation, streptomycin exhibits the largest Zone of Inhibition (ZoI) values across all tested bacteria (Table 3). *Listeria monocytogens* shows the highest sensitivity, with a ZoI of 16.2 mm, followed closely by *Enterococcus faecalis* (16.8 mm), *Salmonella typhi* (15.4 mm), and *Acinetobacter baumannii* (11.8 mm). Streptomycin is highly effective against all the tested bacteria, particularly *E. faecalis*. PFFE-AgNPs displays strong antibacterial activity (Figure 7). It has larger ZoI values compared to PFFE and AgNO_3_ alone. *L. monocytogens* and *E.faecalis* are more sensitive, with ZoI values of 13.5 mm and 14.1 mm, respectively. *S. typhi* (12.8 mm) and *A. baumannii* (8.7 mm) are also inhibited but to a lesser extent. AgNO_3_ shows moderate antibacterial activity, and its ZoI values are notably smaller compared to the streptomycin and PFFE-AgNPs. *E. faecalis* exhibits the highest sensitivity (9.1 mm), followed by *A. baumannii* (8.7 mm), *S. typhi* (7.6 mm), and *L. monocytogens* (8.5 mm). If the concentration of PFFE-AgNPs were to be increased, it might lead to a potential increase in the antibacterial activity against all the tested pathogens. While AgNO_3_ does exhibit a high degree of antibacterial activity, it is important to highlight that its efficacy is notably diminished compared to the robust effects observed with streptomycin and the impressive performance of PFFE-AgNPs. PFFE demonstrates moderate antibacterial activity, as evidenced by the fact that its Zone of Inhibition values are generally lower in comparison to the other tested substances. *L. monocytogens* has the largest ZoI (6.4 mm), while the other bacteria show slightly smaller ZoI values. PFFE alone is not effective as an antibacterial agent. Further, in this test, we have also tested a mixture of PFE and AgNO_3_ before the formation of AgNPs (at 0 h). It displays varied antibacterial activity, and its ZoI values are comparable to PFFE-AgNPs for *L. monocytogens* but notably lower for *E. faecalis*, *S. typhi*, and *A. baumannii*. In the current study, streptomycin demonstrates robust antibacterial effects against all the tested bacteria, surpassing all other the substances examined. PFFE-AgNPs demonstrate strong activity, particularly against *L. monocytogens* and *E. faecalis*. PFFE and AgNO_3_ show promise but with varying effectiveness, while AgNO_3_ displays moderate activity. PFFE alone exhibits moderate antibacterial properties. The effectiveness of these substances varies among the different bacterial species, highlighting the importance of considering specific bacterial strains when assessing antibacterial potential.

In our present experiment, we have detailed the plausible mechanism responsible for the antibacterial activity of PFFE-AgNPs. This elucidation provides insights into the underlying processes contributing to the observed antibacterial effects within the context of our study. PFFE-AgNPs interact with bacterial DNA and causes its fragmentation. As a result, bacterial replication does not occur [80]. PFFE-AgNPs can bind with the catalytic sites of enzymes in bacteria and make them inactive. As a result, bacterial metabolism stops [81]. PFFE-AgNPs bind with important cytosolic or membrane proteins of bacteria, which leads to loss of functional form or the degradation of proteins [81,82,83]. PFFE-AgNPs produces pores on the bacterial cell wall which leads to the leakage of important ions and causes change in the membrane proton gradient. All these actions of PFFE-AgNPs lead to bacterial cell death [82,83,84]. The exact mechanism of antibacterial activity may vary depending on the size, shape, and surface chemistry of the AgNPs as well as the type of bacteria being targeted [85]. However, the overall effect is a reduction in bacterial growth and viability, making AgNPs a promising approach for the development of novel antibacterial agents [82,83,84,85,86]. The antibacterial effects of AgNPs were supported by previous reports. *Ocimum sanctum-* and *Reinwardtia indica*-flavonoid-synthesized AgNPs showed antibacterial activity against Gram-positive and Gram-negative bacterial cultures [66,67]. The AgNPs synthesized using flavonoids showed antibacterial activity against pathogens, including *Pseudomonas aeruginosa*, *Staphylococcus aureus*, and *Escherichia coli* [80,81,82]. 

The selection of bacteria in our study holds significant clinical relevance, as these strains are directly associated with human infections that have substantial public health implications. For instance, *L. monocytogens* and *S. typhi* are well-known pathogens responsible for causing foodborne illnesses, posing a threat to individuals who consume contaminated food [87]. Additionally, *E. faecalis*, a common bacterium, is often implicated in hospital-acquired infections, particularly affecting individuals with compromised immune systems or prolonged hospital stays. Similarly, *A. baumannii*, characterized as an opportunistic pathogen, is frequently linked to hospital-acquired infections, primarily affecting those with compromised immunity or extended hospitalization [87,88]. The assessment of antibacterial activity against these strains is paramount due to the substantial impact they exert on public health. A prevailing concern surrounding these bacteria is their ability to rapidly develop resistance to multiple antibiotics, even those conventionally used to treat bacterial infections. This multidrug resistance phenomenon significantly restricts treatment options, subsequently exacerbating the severity of infections and complicating therapeutic interventions [89]. In our study, we strategically selected bacteria from diverse taxonomic groups, encompassing both Gram-positive strains, such as *L. monocytogens* and *E. faecalis*, as well as Gram-negative strains, including *S. typhi* and *A. baumannii* [87,88,89]. This deliberate choice enables us to gain comprehensive insights into the effectiveness of the tested substances against a broad spectrum of pathogens. 

Based on this study’s findings that PFFE-AgNPs exhibit effective antibacterial activity against a range of pathogens, including clinically relevant strains associated with hospital-based infections, several potential applications emerge for utilizing PFFE-AgNPs in healthcare settings. PFFE-AgNPs’ potent antibacterial properties make them promising candidates for the treatment of hospital-acquired infections. Since PFFE-AgNPs demonstrated effectiveness against pathogens such as *A. baumannii*, a common culprit in healthcare-associated infections, incorporating PFFE-AgNPs into wound dressings, antimicrobial coatings for medical equipment, or even localized treatments could aid in preventing and treating infections that often arise in healthcare environments [90]. AgNPs could be integrated into hospital labware such as petri dishes, sample containers, and laboratory equipment. This incorporation could impart antimicrobial properties to these materials, reducing the risk of contamination and ensuring accurate and reliable test results [91].

Incorporating AgNPs into the fabric of these garments could provide an added layer of protection by inhibiting the growth and spread of bacteria. This could contribute to maintaining a hygienic environment and minimizing the risk of cross-contamination. AgNPs’ antibacterial activity can be harnessed in the design of hospital diagnostic kits. Incorporating these into components of diagnostic kits, such as swabs or culture media, could help ensure the accuracy of test results by reducing the risk of bacterial contamination during sample collection and processing [89,90,91,92]. In the realm of medical device implants, AgNPs hold potential for reducing the risk of infection associated with implantation procedures. By incorporating AgNPs into the surface of implantable devices or coatings, the growth of bacteria around the implant site could be inhibited, promoting successful integration and reducing the likelihood of post-surgical infections [91,92]. The study’s results, indicating the broad-spectrum antibacterial activity of AgNPs, including their efficacy against Gram-positive and Gram-negative strains, provide a strong foundation for exploring these various applications. However, it is important to note that while AgNPs show promise, further research and careful consideration of their potential cytotoxicity and environmental impact are necessary before their widespread implementation in healthcare settings.

The utilization of PFFE-AgNPs offers several distinct advantages over conventional methods and materials, which contribute to their potential as a promising antimicrobial agent: 

Enhanced Antibacterial Activity: The incorporation of AgNPs derived from PFFE amplifies the inherent antibacterial properties of both components. This synergistic effect results in heightened antibacterial activity, potentially surpassing the individual contributions of AgNO_3_ and PFFE. This enhanced activity can lead to improved efficacy in combating bacterial infections [93].

Biocompatibility: PFFE, being a natural plant extract, is likely to possess inherent biocompatibility and reduced cytotoxicity compared to synthetic compounds. This makes PFFE-AgNPs a safer option for medical applications, as they may exhibit lower toxicity toward mammalian cells while retaining potent antibacterial effects [34,94].

Selectivity: PFFE-AgNPs can potentially exhibit selective antibacterial activity, targeting harmful bacteria while sparing beneficial microorganisms. This selectivity can contribute to the preservation of the body’s natural microbial balance, reducing the risk of microbial imbalances or dysbiosis [95].

Multi-Target Action: PFFE-AgNPs can offer a multi-targeted approach to combating bacterial infections. The nanoparticles can act through various mechanisms, such as disrupting cell membranes, interfering with bacterial enzymes, and generating reactive oxygen species, thereby reducing the likelihood of bacterial resistance development [96]. 

Long-lasting Effects: The stability and sustained release properties of PFFE-AgNPs may provide prolonged antibacterial effects. This sustained action can reduce the need for frequent dosing and improve patient compliance [97].

Combating Antibiotic Resistance: Given the rising concern of antibiotic-resistant bacteria, PFFE-AgNPs offer an alternative strategy for combating infections that may be less prone to resistance development due to their multifaceted mode of action [94,95,96,97].

Potential Synergy with Other Therapies: PFFE-AgNPs’ diverse properties, including their cytotoxic effects against cancer cells and antioxidant activity, suggest potential synergy with other therapies. This opens doors for combination treatments that address multiple aspects of disease progression [98].

Versatility: The biocompatibility and multi-functionality of PFFE-AgNPs opens doors to diverse biomedical applications beyond antibacterial use. Their potential ranges from cancer therapy to wound healing and targeted drug delivery, expanding their utility in various medical contexts [94,95,96,97,98].

Natural Source: Utilizing plant-derived extracts, such as PFFE, aligns with the trend toward natural and holistic approaches in medicine. This natural origin may resonate with patients seeking treatments that more natural [92,93,94,95,96,97,98].

In summary, PFFE-AgNPs offer a unique amalgamation of potent antibacterial activity, biocompatibility, sustainable synthesis, and potential versatility. These advantages position PFFE-AgNPs as a promising candidate for addressing bacterial infections while mitigating some of the limitations associated with conventional antimicrobial agents. However, rigorous research is essential to confirm these advantages and ensure their safety and efficacy in real-world biomedical applications.

Antimicrobial agents play a critical role in modern medicine by helping to combat a wide range of microbial infections caused by bacteria, viruses, fungi, and other pathogens. The rise of drug-resistant microbes, commonly referred to as antibiotic-resistant or multidrug-resistant organisms, has highlighted the urgent need for new and innovative approaches to tackling infectious diseases. As traditional antibiotics and antimicrobial drugs become less effective against resistant strains of microbes, researchers are exploring a diverse array of compounds to find novel solutions. By exploring and harnessing the properties of natural compounds, including organic derivatives [99], bioorganic moieties [100], synthetics of bioactive moieties [101] silver-based copolymers [102] and metal-organic frameworks [103], scientists aim to discover new strategies to effectively combat microbial infections and reduce the impact of drug resistance. These efforts hold the promise of revolutionizing the field of antimicrobial therapy and improving public health outcomes.

### 2.7. Antioxidant Activity of the PFFE-AgNPs

The antioxidant activity of the PFFE-AgNPs was evaluated using in vitro DPPH and H_2_O_2_ radical scavenging assays. All the tests were conducted three times. The average or mean values were represented with the standard deviation (Mean ± SD). In the DPPH radical scavenging assay, we observed the dose-dependent activity of PFFE-AgNPs against DPPH radicals. This means that as the concentration of PFFE-AgNPs increased, the percentage inhibition of DPPH radicals also increased. The results are presented in Figure 8a, which shows a graph depicting the relationship between PFFE-AgNPs concentration and the percentage inhibition of DPPH radicals. It was noted that at the highest concentration of PFFE-AgNPs, the maximum inhibition achieved was 58.96%.

To determine the effectiveness of the PFFE-AgNPs as radical scavengers, we also calculated the IC_50_ value, which represents the concentration of the substance required to scavenge 50% of the radicals. In this study, the IC_50_ value for the PFFE-AgNPs in the DPPH radical scavenging assay was found to be 72.81 µg/mL. A lower IC_50_ value indicates a higher antioxidant activity, suggesting that the PFFE-AgNPs were effective at scavenging DPPH radicals.

Similarly, the antioxidant activity of the PFFE-AgNPs was evaluated using the H_2_O_2_ radical scavenging assay. The researchers observed the concentration-dependent inhibition of H_2_O_2_ radicals by the PFFE-AgNPs, as shown in Figure 8b. The IC_50_ value for the PFFE-AgNPs in this assay was determined to be 92.48 µg/mL. This indicates that the PFFE-AgNPs exhibited effective scavenging of H_2_O_2_ radicals.

In summary, the study demonstrated that the biosynthesized PFFE-AgNPs exhibited dose-dependent antioxidant activity as assessed using the DPPH and H_2_O_2_ radical scavenging assays. The PFFE-AgNPs showed effective inhibition of DPPH and H_2_O_2_ radicals, as indicated by the percentage inhibition and IC_50_ values obtained.

The effective antioxidant activity of PFFE-AgNPs might be due to flavonoids participating in the bioreduction and capping of PFFE-AgNPs. Oxidative stress refers to an imbalance between the production of reactive oxygen species (ROS) and the body’s ability to detoxify and neutralize their harmful effects. ROS are highly reactive molecules that can damage cellular components such as lipids, proteins, and DNA, leading to cell dysfunction and death. Antioxidants are compounds that can neutralize ROS and prevent oxidative damage. AgNPs act as free radical scavengers by donating electrons to neutralize the unpaired electrons of the free radicals, which helps prevent the cellular damage caused by oxidative stress. Additionally, AgNPs have been found to enhance the activities of antioxidant enzymes, such as superoxide dismutase (SOD) and catalase (CAT), which further aids in the elimination of free radicals and ROS. AgNPs also have the ability to chelate metal ions, which can contribute to their antioxidant activity. Chelation involves binding metal ions to the surface of AgNPs, which reduces the availability of the ions to participate in the generation of free radicals and ROS. Furthermore, AgNPs have been found to inhibit lipid peroxidation, which is a process that leads to the generation of free radicals and can result in cellular damage. AgNPs can prevent lipid peroxidation by inhibiting the production of reactive aldehydes and other lipid peroxidation products. Overall, the antioxidant activity of AgNPs is mediated by a combination of mechanisms, including free radical scavenging, the enhancement of antioxidant enzyme activities, metal chelation, and the inhibition of lipid peroxidation. These properties make AgNPs a promising candidate for various biomedical applications, including antioxidant therapy. However, the potential health risks associated with AgNPs, including their potential to induce oxidative stress, are still under investigation.

### 2.8. Cytotoxicity Activity of the PFFE-AgNPs

In the scope of our study, we meticulously assessed the cytotoxicity activity of PFFE-AgNPs against human colon carcinoma (COLO205) and mouse melanoma (B16F10) cells. To provide a meaningful context for our findings, we compared the effects of PFFE-AgNPs with those of the well-established FDA-approved drug, doxorubicin, which is widely acknowledged for its potent anticancer properties. Doxorubicin demonstrated remarkable efficacy, exhibiting IC_50_ values of 1.28 µM against B16F10 cells and 2.12 µM against COLO205 cells. In stark contrast, PFFE alone exhibited significantly lower anticancer effects, with IC_50_ values of 235.8 μg/mL against B16F10 cells and 285.7 μg/mL against COLO205 cells. This discrepancy underscores the modest anticancer potential of PFFE in its isolated form. However, with the introduction of PFFE-AgNPs, a notable shift was observed. The concentration-dependent decrease in cell viability of both COLO205 and B16F10 cells in response to PFFE-AgNPs was evident from our findings. The escalation of the PFFE-AgNPs dosage from 25 to 100 μg/mL exhibited a substantial reduction in cell viability, signifying a dose-dependent cytotoxic effect (Figure 9). Remarkably, the subsequent increase in dosage from 100 to 200 μg/mL did not lead to a proportionate escalation in inhibition, indicating a potential saturation point. Notably, PFFE-AgNPs demonstrated a significant maximum inhibition of 85.2% against COLO205 cells and 80.9% against B16F10 cells, substantiating their robust cytotoxic activity. This is further underscored by the IC_50_ concentrations of PFFE-AgNPs against COLO205 and B16F10 cells, measured at 59.57 μg/mL and 69.33 μg/mL, respectively. Such outcomes validate the substantial anticancer potential of PFFE-AgNPs against these cancer cell lines. The literature supports the notion of AgNPs as promising anticancer agents against various cancer cell lines, including COLO205 and B16F10. This is attributed to their distinctive physicochemical properties that enable them to target cancer cells effectively while minimizing harm to healthy cells.

In envisioning the development of PFFE-AgNPs as potent anticancer agents, a strategic approach could involve synergizing their action with doxorubicin. The combined administration of doxorubicin and AgNPs holds the potential for enhanced therapeutic efficacy through complementary mechanisms of action. Doxorubicin’s established ability to inhibit DNA replication and AgNPs’ distinctive properties that induce cellular stress and apoptosis could potentially result in a more profound anticancer effect.

Looking towards the future, the amalgamation of drugs and nanoparticles represents a promising avenue in the realm of anticancer therapy. The synergy between pharmacological agents and nanoparticles offers the prospect of heightened therapeutic outcomes while concurrently reducing the adverse effects often associated with conventional chemotherapy. The controlled and targeted delivery enabled by nanoparticles could facilitate the selective accumulation of drugs within cancer cells, thereby enhancing treatment efficacy and minimizing damage to healthy tissues. This innovative approach holds the promise of revolutionizing cancer treatment paradigms and advancing patient care.

The results from our study underscore the substantial cytotoxic potential of PFFE-AgNPs against COLO205 and B16F10 cells when compared with both PFFE alone and the reference drug doxorubicin. The emergence of AgNPs as a viable and potent anticancer agent, coupled with the prospect of combining them with established drugs, opens up exciting avenues for the development of future anticancer therapies with enhanced efficacy and reduced side effects.

The exact mechanism by which AgNPs exert their anticancer activity is not completely understood, but several studies have proposed some possible mechanisms. One of the proposed mechanisms is that AgNPs induce apoptosis, or programmed cell death, in cancer cells. This process is mediated by the activation of caspases, a family of protease enzymes that play a crucial role in the execution of apoptosis [104]. AgNPs have been shown to increase the expression and activation of caspases in cancer cells, leading to their death. Another proposed mechanism is that AgNPs induce oxidative stress in cancer cells [105]. AgNPs have been shown to increase ROS production in cancer cells, which leads to damage to their cellular components such as DNA, proteins, and lipids, ultimately leading to their death [106]. Additionally, AgNPs have been shown to inhibit cancer cell proliferation by disrupting the cell cycle. This disruption is mediated by the downregulation of cyclin-dependent kinases (CDKs), which are enzymes that play a crucial role in regulating the cell cycle. AgNPs have been shown to inhibit CDKs activity, leading to cell cycle arrest and the subsequent death of cancer cells [107]. Furthermore, AgNPs have been shown to exhibit antiangiogenic properties, which means that they can inhibit the growth of new blood vessels that are necessary for tumor growth and metastasis. This mechanism is mediated by the inhibition of the vascular endothelial growth factor (VEGF), a protein that promotes angiogenesis [108]. Overall, the anticancer activity of AgNPs against COLO205 and B16F10 cell lines is likely due to a combination of the mechanisms described above [104,105,106,107,108]. However, further research is needed to fully elucidate the mechanisms and to optimize the use of AgNPs as an anticancer agent.

The novelty of this research lies in its innovative approach to synthesizing AgNPs through biomimetic synthesis, specifically utilizing flavonoids extracted from *P. frutescens*. While the synthesis of AgNPs is not a new concept, the method employed in this study deviates from conventional chemical approaches. Instead, it draws inspiration from nature’s intricate processes, mimicking the way biomolecules interact to craft nanoparticles with precision.

Traditionally, chemical methods for the synthesis of AgNPs often involves the use of harsh reducing agents or stabilizers that can introduce toxicity concerns and limit their applicability in sensitive biological systems. In contrast, the biomimetic approach embraced in this research capitalizes on the inherent properties of flavonoids present in *P. frutescens* extract. These natural compounds act as both reducing and stabilizing agents, enabling the synthesis of AgNPs in a more eco-friendly and biocompatible manner.

Furthermore, the utilization of *P. frutescens* as a source of flavonoids introduces a unique dimension to the research. *P. frutescens* is known for its rich repository of bioactive compounds, particularly flavonoids, which have demonstrated various therapeutic properties, including antioxidant, anti-inflammatory, and antimicrobial activities. Incorporating these bioactive compounds into the synthesis process not only offers a green and sustainable alternative to traditional methods but also introduces the potential for imbuing the AgNPs with additional functionalities.

The integration of flavonoids from *P. frutescens* into the biomimetic synthesis of AgNPs opens doors to a plethora of exciting possibilities. By exploiting the inherent affinity of flavonoids for metal ions, the research aims to achieve precise control over the size, shape, and stability of the nanoparticles. This level of control is often challenging to attain using conventional chemical approaches, which can lead to a broader distribution of particle sizes and shapes.

Moreover, the multifaceted properties of flavonoids, such as their antioxidant and antibacterial attributes, hold the promise of conferring these beneficial traits onto the synthesized AgNPs. This potential for multifunctionality is a significant departure from traditional chemical synthesis methods, in which the incorporation of bioactive properties is often complex and limited.

In essence, the novelty of this research emerges from the synergistic combination of biomimetic synthesis, flavonoids from *P. frutescens*, and the inherent advantages of green synthesis. By harnessing these elements, the study not only contributes to the advancement of nanobiotechnology but also holds the potential to pave the way for a new generation of AgNPs with enhanced biocompatibility, multifunctionality, and application in diverse fields, ranging from medicine to environmental remediation.

While the findings suggest the potential of PFFE-AgNPs as a versatile nanomaterial with applications in cancer therapy and infection control, further research is imperative. In particular, future investigations should delve into the safety and efficacy of PFFE-AgNPs in vivo and explore their potential contributions to diverse areas of medicine. This research opens doors to innovative advancements in the realm of nanomedicine, beckoning towards a brighter future for biomedical applications.

## 3. Materials and Methods

### 3.1. Preparation of PFFE

Solvent extraction is the most used method to extract flavonoids from *P. frutescens* because of its simplicity, low cost, and high efficiency. Perilla leaves are harvested, washed, and dried. The dried leaves are ground into a fine powder using a grinder or blender. Common solvents used for flavonoid extraction include methanol, ethanol, and water. Methanol is the most efficient solvent for flavonoid extraction from perilla leaves, but ethanol and water can also be used. The powdered perilla leaves are mixed with the methanol (1:5) and left to stand for several hours or overnight. The mixture is then filtered, and the filtrate is collected. The collected filtrate is concentrated using a rotary evaporator to remove the solvent and obtain the flavonoid extract. This methanolic extract was subjected to drying using rotavapor to form a solid residue, which was dissolved in ethyl acetate. The ethyl acetate-soluble fraction was subjected to purification over a silica gel column using a step gradient of methanol-ethyl acetate (1:1 to 3:7) to yield a flavonoid extract, which was verified using UV-Vis spectroscopy between 240 and 400 nm [109].

### 3.2. Estimation of Total Flavonoid Content (TFC)

TFC of the PFFE was determined using an aluminum chloride colorimetric method [13]. In this method, 1mg/mL of PFFE was added to a 10 mL volumetric flask containing 4 mL of sterilized double distilled water (SDDW). At zero time, 0.3 mL of 10% AlCl_3_.6H_2_O and 0.3 mL of 5% NaNO_2_ were added. After 5 min, 2 mL of 1 M NaOH was added to the mixture and the final volume was made up to 10 mL with SDDW. The solution was mixed well, and the absorbance was read at 510 nm. The TFC was determined using a quercetin standard curve. TFC was expressed as mg of quercetin equivalents (QE) per gram of dried sample. 

### 3.3. Biofabrication of PFFE-AgNPs

Two mM AgNO_3_ solution was prepared by the addition of 339 mg of AgNO_3_ in 1 L of SDDW. A total of 95 mL of 2 mM AgNO_3_ was added to 5 mg/mL of PFFE. This mixture was boiled for 1 h between 55 and 60 °C, cooled in a dark chamber for 30 min, and then observed for color change in the solution from light yellow to dark brown [110]. The color change indicates the formation of AgNPs in the colloidal solution.

### 3.4. Characterization of PFFE-AgNPs

#### 3.4.1. UV-Vis Analysis of PFFE-AgNPs

Initially, the color change in the reaction solution from light yellow to dark brown indicates the synthesis of PFFE-AgNPs. Furthermore, the synthesis of PFFE-AgNPs was confirmed using a ultraviolet–visible (UV-Vis) spectrum analysis between 200 and 800 nm (Waltham, Massachusetts, USA) [110].

#### 3.4.2. FTIR Analysis of PFFE-AgNPs

The solution of PFFE-AgNPs was purified by centrifugation to remove the unbounded plant molecules. Centrifugation was carried out and repeated three times at 15,000 rpm for 15 min. The pellet was collected and dried into pure powder. This pure powder of PFFE-AgNPs was used for further studies. To detect the functional groups involved in the synthesis and capping of PFFE-AgNPs, a Fourier transform infrared (FTIR) spectrum (Alpha interferometer, Bruker, Karlsruhe, Germany) was recorded between 500 and 4000 cm^−1^ with a resolution of 2 cm^−1^ [111].

#### 3.4.3. XRD Analysis of PFFE-AgNPs

X-ray diffraction (XRD) analysis of PFFE-AgNPs was carried out to ascertain the crystal structure in a scanning range of 10–90° with a step size of 2θ. The XRD pattern for the crystal structure of PFFE-AgNPs was recorded (Bruker AXS GmbH, Karlsruhe, Germany) [112].

#### 3.4.4. TEM Analysis of PFFE-AgNPs

The size, morphology, and selected area electron diffraction (SAED) of PFFE-AgNPs were analyzed using transmission electron microscopy (JEM-2100 Plus Electron Microscope, JEOL Ltd., Tokyo, Japan) [113].

#### 3.4.5. DLS Analysis of PFFE-AgNPs

Twenty µg of the dried powder of PFFE-AgNPs was dispersed in 20 mL of Milli-Q water, and this solution was used for DLS (dynamic light scattering) studies. DLS studies were carried out to determine the particle size distribution, polydispersity index (PDI), and zeta potential measurement (Brookhaven instruments, Holtsville, NY, USA) [13].

### 3.5. Antibacterial Activity of PFFE-AgNPs

The antibacterial activity of the PFFE-AgNPs was evaluated using a disc diffusion assay [114] against Gram-positive (*Listeria monocytogens* and *Enterococcus faecalis*) and Gram-negative (*Salmonella typhi* and *Acinetobacter baumannii*) bacteria pathogens. A total of 150 µL of each bacterial culture was spread on the nutrient agar media. Then, three sterile paper discs containing test samples were placed on the nutrient agar media. Streptomycin, PFFE, AgNO_3,_ and PFFE-AgNPs were used as test samples at a concentration of 25 µL per each disc. The nutrient agar media plates were incubated for 24 h in the incubator at 37 °C. Inhibition zones were observed and measured after 24 h. 

### 3.6. Antioxidant Activity of the PFFE-AgNPs

The in vitro antioxidant activity of the biosynthesized PFFE-AgNPs was checked using 1,1-diphenyl-2-picrylhydrazyl (DPPH) and hydrogen peroxide (H_2_O_2_) radical scavenging assays [115]. The standard antioxidant ascorbic acid was used as a positive control. In DPPH assay, different concentrations (25, 50, 75, and 100 μg/mL) of samples were added to methanol and amounted a final volume of 1 mL. To this, 2 mL of DPPH stock solution (1 mM/L prepared in methanol) was added and incubated for 1 h in the dark at room temperature. After an hour of incubation, the absorption values were recorded at 517 nm. In the H_2_O_2_ assay, different concentrations (25, 50, 75, and 100 μg/mL) of samples were added to 2 mL of H_2_O_2_ solution (prepared in a 40 mM phosphate buffer (pH 7.4)) and then incubated for 10 min. After incubation, the absorbance was recorded at 230 nm. DPPH/H_2_O_2_ scavenging activity was calculated using the following formula: % Inhibition = [(Ac − As)/Ac] × 100, where Ac is the absorbance of the control and As is the absorbance of the sample. 

### 3.7. Cytotoxicity Activity of the PFFE-AgNPs

In this study, the anticancer activity of PFFE-AgNPs was evaluated using MTT [3-(4,5-dimethylthiazol-2-yl)-2,5-diphenyl tetrazolium bromide] assay [116] against human colon cancer (COLO205) and mouse melanoma (B16F10). The cells were suspended in 100 µL of medium in each well of 96 well plates and incubated in a 5% CO_2_ incubator at 37 °C overnight. Then, 100 µL of PFFE-AgNPs (25, 50, 75, 100, 150, and 200 µg/mL) were added to the cell suspension and incubated for 24 h. Then, MTT (10 μL) was added and incubated for 3 h. Then, the centrifugation of the 96 well plates was carried out for 10 min. After centrifugation, the medium was removed, and the formazan blue crystals were melted in dimethyl sulfoxide. Then, the absorbance values were recorded at 570 nm, and the inhibition of cell viability was calculated. The median inhibition concentration (IC_50_) values were determined.

## 4. Conclusions

In conclusion, the present study demonstrated that PFFE extract from *P. frutescens* can act as a reducing and capping agent for the biosynthesis of AgNPs, resulting in the formation of stable, monodisperse, and negatively charged spherical nanoparticles with a face-centered cubic crystal lattice structure. These PFFE-AgNPs exhibited promising biomedical applications, including significant antibacterial activity against both Gram-positive and Gram-negative bacteria, antioxidant activity against DPPH and H_2_O_2_ free radicals, and cytotoxic effects against cancer cell lines. The antibacterial activity of PFFE-AgNPs is of particular importance, as antibiotic resistance is a growing global health concern. The antioxidant activity of PFFE-AgNPs also has potential for therapeutic applications in oxidative stress-related diseases. Moreover, the cytotoxic effects of PFFE-AgNPs against cancer cell lines suggest their potential for use in anticancer therapies. Overall, this study highlights the potential of using plant extracts as a green and sustainable approach for the synthesis of AgNPs with biomedical applications. Further research is needed to fully understand the mechanisms of action of PFFE-AgNPs and their potential toxicity to healthy cells before their transition into clinical applications.

## Figures and Tables

**Figure 1 molecules-28-06431-f001:**
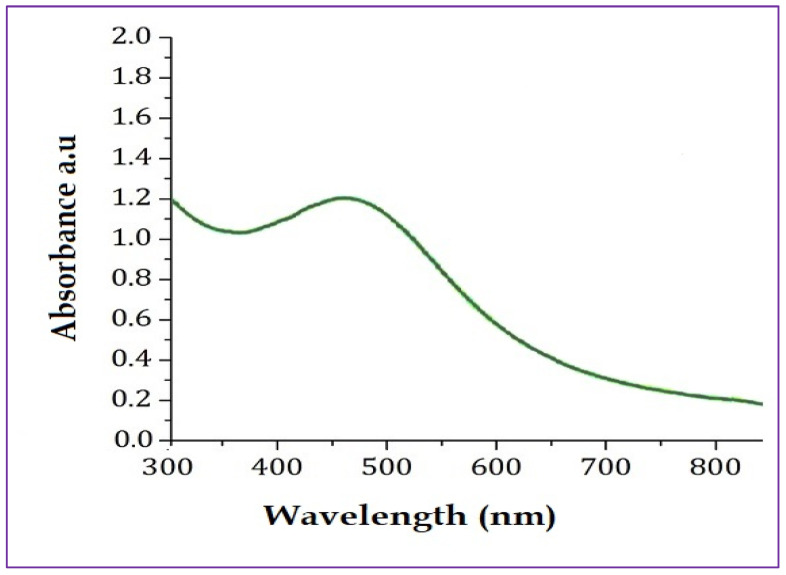
UV-Vis analysis of PFFE-AgNPs showed surface plasmon resonance peak at 440 nm.

**Figure 2 molecules-28-06431-f002:**
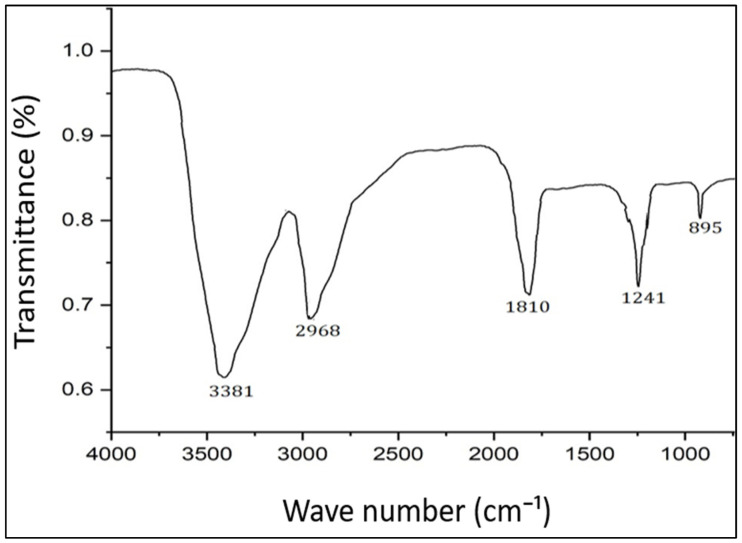
FTIR spectrum of PFFE-AgNPs.

**Figure 3 molecules-28-06431-f003:**
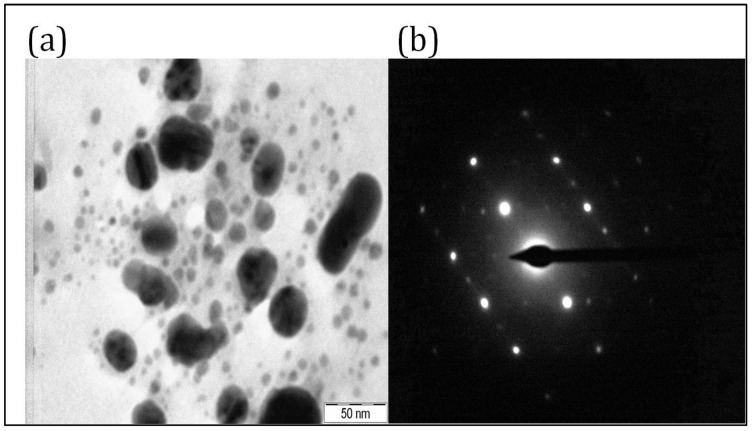
(**a**) TEM micrograph at 50 nm scale and (**b**) SAED pattern of PFFE-AgNPs.

**Figure 4 molecules-28-06431-f004:**
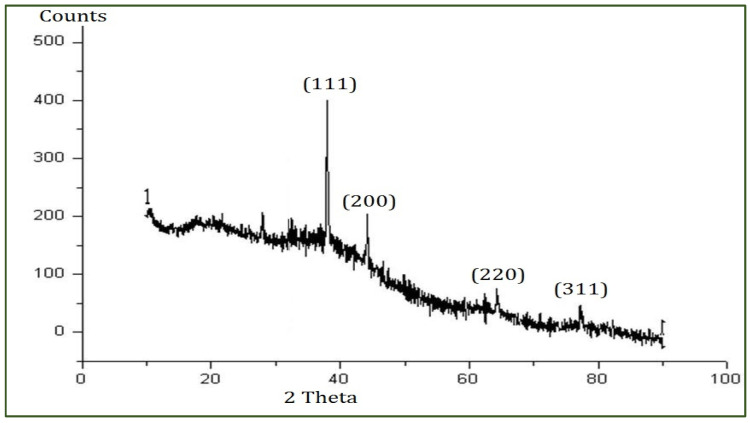
XRD pattern of PFFE-AgNPs showing diffraction peaks.

**Figure 5 molecules-28-06431-f005:**
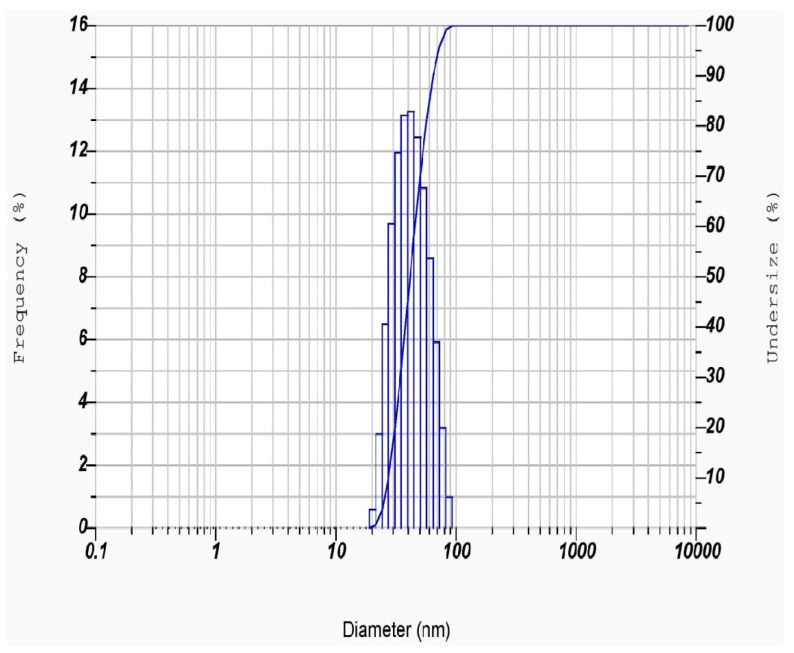
Particle size distribution of PFFE-AgNPs.

**Figure 6 molecules-28-06431-f006:**
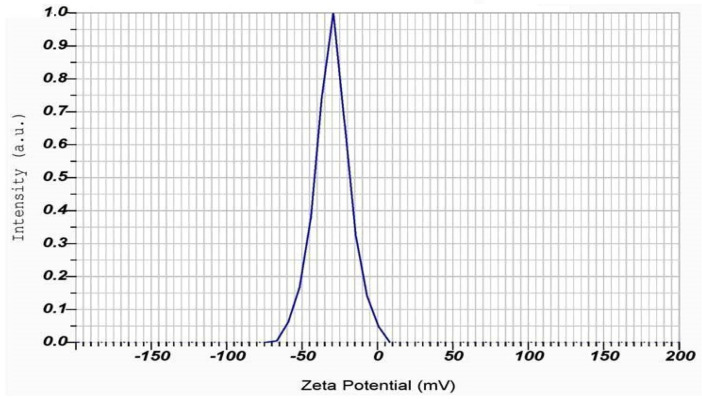
Zeta potential measurement of PFFE-AgNPs.

**Figure 7 molecules-28-06431-f007:**
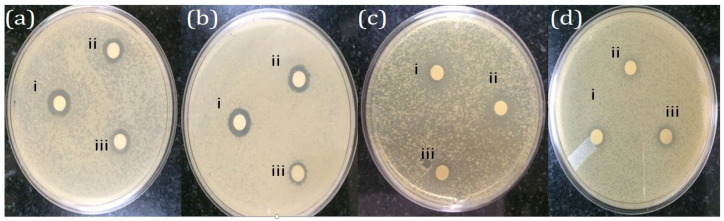
Antibacterial activity of PFFE-AgNPs against (**a**) *Listeria monocytogens*, (**b**) *Enterococcus faecalis*, (**c**) *Salmonella typhi*, and (**d**) *Acetobacter boumani*; i—indicates Streptomycin; ii—indicates PFFE-AgNPs; and iii—indicates AgNO_3_.

**Figure 8 molecules-28-06431-f008:**
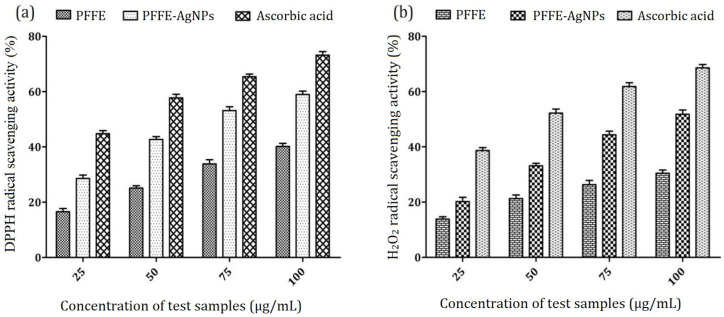
(**a**) DPPH scavenging activity (**b**) H_2_O_2_ scavenging activity of PFFE-AgNPs. All the results were represented as the mean ± SD of three replicates.

**Figure 9 molecules-28-06431-f009:**
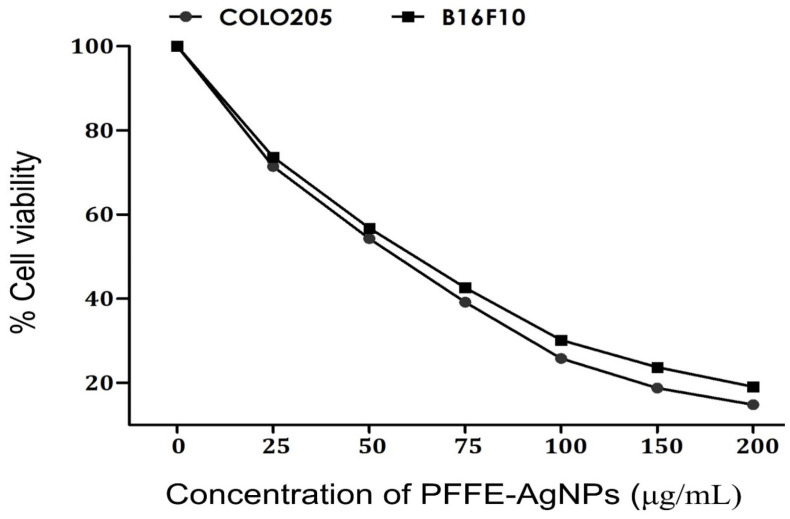
Anticancer activity of PFFE-AgNPs against human colon carcinoma (COLO205) and mouse melanoma (B16F10). All the results were represented as the mean ± SD of three replicates.

**Table 1 molecules-28-06431-t001:** Particle size distribution of PFFE-AgNPs.

Peak No.	S.P. Area Ratio	Mean	S. D.	Mode
1	1.00	44.0 nm	14.4 nm	41.9 nm
2	---	--- nm	--- nm	--- nm
3	---	--- nm	--- nm	--- nm
Total	1.00	44.0 nm	14.4 nm	41.9 nm
Z-Average	3489.3 nm			
PI	0.321			

**Table 2 molecules-28-06431-t002:** Zeta potential analysis of PFFE-AgNPs.

Peak No.	Zeta Potential	Electrophoretic Mobility
1	−30.0 mV	−0.000233 cm^2^/Vs
2	--- mV	--- cm^2^/Vs
3	--- mV	--- cm^2^/Vs

**Table 3 molecules-28-06431-t003:** Antibacterial activity of PFFE-AgNPs.

Substance	*Listeria monocytogens*	*Enterococcus faecalis*	*Salmonella typhi*	*Acetobacter boumani*
Streptomycin	16.2 mm	16.8 mm	15.4 mm	11.8 mm
PFFE-AgNPs (PFFE and silver nanoparticles)	13.5 mm	14.1 mm	12.8 mm	8.7 mm
PFFE	6.4 mm	6.1 mm	5.8 mm	5.1 mm
PFFE and AgNO_3_ (before incubation)	13.1 mm	8.6 mm	6.8 mm	6.5 mm
AgNO_3_ (silver nitrate)	8.5 mm	9.1 mm	7.6 mm	8.7 mm

## Data Availability

Not Applicable.

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
