# Peer review of "Exploring the Biomedical Applications of Biosynthesized Silver Nanoparticles Using Perilla frutescens Flavonoid Extract: Antibacterial, Antioxidant, and Cell Toxicity Properties against Colon Cancer Cells"

_molecules, 2023, doi:10.3390/molecules28176431_

Round 1

Reviewer 1 Report

The manuscript reported the biomedical applications of P. frutescens flavonoid extract silver nanoparticles on anticancer, antibacterial, and antioxidant properties. As mentioned by authors say, PFFE-AgNPs have potential as a multi-functional nanomaterial for biomedical applications, particularly in cancer therapy and infection control.

  Since the biofabrication, characterization and activities evaluation of silver nanoparticles have already been wisely researched, the innovativeness of the manuscript is very limited. In addition, they are many mistakes on English grammar, data analysis, Figures notes, and so on. Therefore after a long period of consideration it is with regret that I recommend rejection.

Specific performance in the following:

1.The introduction is superficial and lacks in depth. A description or review of current study of P. frutescens flavonoid extract silver nanoparticles will be very helpful to the significant paper. In particular, discussing what are the situation and challenges for now.

2. More detail is required in the “Materials and Methods” section to enable anyone to be able to undertake this study in the way that it was originally undertaken, such as the analysis of X-ray diffraction (XRD)   (scanning range ??? step size ???), etc.

3. Fig. 2: The first letter of the X,Y-axis label should be capitalized.

4. Fig. 8, Fig. 9: There is no significant description in Figure and results, such as Tukey's test, and so on.

5. The first letter of the X,Y-axis label should be capitalized.

6. The current manuscript have many grammatical errors, therefore the paper could benefit from proof reading/ language editing. Line 31 on page 2: the “CO2” should be revise into “CO2”. Line 187 on page 9: the “in vivo” should be italic.…..

7. Referencing has to be improved. The authors reference on general knowledge, the literature survey has to be strengthened and go into detail.

8. The full name of “P. frutescens” should be provided in the title (line 1).

9. The description “PRC.” should be revised to "P.R. China".

There are many mistakes on English grammar.

Author Response

Reviewer 1

Comments and Suggestions for Authors

The manuscript reported the biomedical applications of P. frutescens flavonoid extract silver nanoparticles on anticancer, antibacterial, and antioxidant properties. As mentioned by authors say, PFFE-AgNPs have potential as a multi-functional nanomaterial for biomedical applications, particularly in cancer therapy and infection control.

  Since the biofabrication, characterization and activities evaluation of silver nanoparticles have already been wisely researched, the innovativeness of the manuscript is very limited. In addition, they are many mistakes on English grammar, data analysis, Figures notes, and so on.

Specific performance in the following:

1.The introduction is superficial and lacks in depth. A description or review of current study of P. frutescens flavonoid extract silver nanoparticles will be very helpful to the significant paper. In particular, discussing what are the situation and challenges for now.

The introduction provides an in-depth exploration of the diverse field of nanobiotechnology, which integrates biology, chemistry, and physics to address a wide range of applications. It delves into the remarkable properties and potential applications of metal nanoparticles, highlighting their distinct characteristics originating from surface energy and a significant surface area-to-volume ratio.

The focal point of this introduction is the elucidation of bottom-up approaches to nanoparticle synthesis, with a special emphasis on biomimetic methods inspired by natural processes. This strategy capitalizes on biomolecules to meticulously engineer nanoparticles with exceptional qualities. Within the context of environmentally responsible practices, the concept of green synthesis is elaborated upon. This involves a spotlight on the advantages of utilizing plant extracts in contrast to chemical agents. These natural constituents enhance both biocompatibility and biodegradability, seamlessly aligning with sustainable principles and the development of innovative drug delivery systems.

The introduction also underscores the pivotal role of flavonoids, notable bioactive compounds, in the biomimetic synthesis of silver nanoparticles. The exploration specifically centers on P. frutescens as a potent source of these bioactive compounds. The ultimate objective of this study is to harness the potential of flavonoids derived from P. frutescens for the biomimetic synthesis of silver nanoparticles. This endeavor is founded upon the intrinsic chemical attributes and therapeutic promise of these compounds, aiming to contribute to advancements in this dynamic field.

The present study involving P. frutescens extract in the synthesis of AgNPs aim to explore the green synthesis approach, which involves using natural sources instead of conventional chemical methods. Green synthesis offers several advantages, including eco-friendliness, cost-effectiveness, and the potential for producing nanoparticles with unique properties.

One of the significant challenges in this area of research is the optimization of synthesis conditions. The extraction process and subsequent reduction of silver ions to form nanoparticles require careful control of various parameters such as, temperature, concentration, and reaction time. Finding the optimal conditions to obtain well-defined nanoparticles with desirable properties can be a complex and time-consuming process.

Please refer from page Number 6 to 7(Revised Manuscript, Introduction section, Page number 6-7, Green color Letters)

Active ingredients and plant extracts are often natural and biocompatible, making them ideal candidates for use in biomedical applications [48]. When utilized in nanoparticle synthesis, these natural compounds can improve the biocompatibility of the nanoparticles, reducing the risk of toxicity and adverse reactions in medical or biological settings [49]. Active ingredients and plant extracts contain various bioactive compounds that can endow nanoparticles with unique functionalities [20, 48-50]. These functionalities can include antibacterial, antifungal, antioxidant, anti-inflammatory, and other therapeutic properties, depending on the specific plant extracts used. Such functionalized nanoparticles have promising applications in medicine, agriculture, and environmental remediation [51]. Combining different active ingredients or plant extracts in nanoparticle synthesis can lead to synergistic effects, where the resulting nanoparticles exhibit enhanced or novel properties compared to individual components [52]. Such synergies may improve the overall performance and efficiency of nanoparticles in various applications [53]. The biodegradability of active ingredients and plant extracts is a desirable property when considering potential biomedical applications of nanoparticles [54]. Biodegradable nanoparticles can be designed to release their payload in a controlled manner, which is particularly advantageous for drug delivery systems. By incorporating active ingredients and plant extracts, researchers can opt for a more sustainable and eco-friendly approach to nanoparticle synthesis [55]. Further plant extracts are abundantly available, making them cost-effective options for nanoparticle synthesis. By using readily accessible natural sources, researchers can reduce production costs and increase the feasibility of large-scale nanoparticle manufacturing [56].

In the context of AgNPs, recent reports have highlighted the successful use of various plant extracts, such as ginger [50], curcumin [57], garlic extract [58], Allium cepa peel extract [59], Aloe vera [60],  green tea [61], seed extracts Brassica oleraceae, B. campestris and B. rapa [62], and Teucrium polium leaf extract [63] as potent reducing agents for the synthesis of AgNPs. These natural extracts contain bioactive compounds, such as polyphenols, flavonoids, and terpenoids, which play a crucial role in the reduction and stabilization of nanoparticles.

Flavonoids are a class of polyphenolic compounds that are widely distributed in the plant kingdom and have a wide range of biological activities, including antioxidant, anti-inflammatory, and antibacterial properties [64]. Flavonoids are used in the synthesis of AgNPs because they have unique chemical properties that make them ideal for reducing silver ions into nanoparticles. Flavonoids are known to have a high affinity for metal ions, and they can act as reducing agents in the presence of metal ions to form nanoparticles [65].

During the biosynthesis of AgNPs, flavonoids present in the plant extract act as reducing agents that help in the conversion of silver ions to AgNPs. Flavonoids contain multiple hydroxyl groups that are capable of reducing metal ions to nanoparticles by donating electrons. The hydroxyl groups present in the flavonoids interact with the silver ions, leading to the reduction of the silver ions and the formation of AgNPs [65, 66]. Furthermore, flavonoids also act as stabilizing agents for AgNPs, preventing their agglomeration and ensuring their stability in solution [65-67]. The hydroxyl groups present in the flavonoids interact with the surface of AgNPs, forming a layer that stabilizes the nanoparticles and prevents them from aggregating. This makes flavonoids a useful tool in the biosynthesis of AgNPs.

In the revised Manuscript, Introduction section is very large contains 1950 words and 70 citations.

  1. Introduction

Nanobiotechnology, an interdisciplinary field that merges biology, chemistry, and physics, has recently emerged as a promising discipline with a wide range of applications [1]. One of the key areas of nanobiotechnology is the synthesis and characterization of nanoparticles, which are particles with sizes below 100 nanometers. Two major approaches for synthesizing nanoparticles are the top-down and bottom-up approaches, with the latter becoming increasingly popular due to its simplicity, cost-effectiveness, and robustness [2]. Metal nanoparticles possess unique properties such as optical, magnetic, catalytic, mechanical, electronic, and thermal properties, which stem from their surface energy, spatial confinement, and high surface area-to-volume ratio [3,4]. The localized surface plasmon resonance (SPR) and surface-enhanced Raman scattering (SERS) of metal nanoparticles make them highly attractive for use in next-generation electronic and biochemical sensors [5,6]. In this context, the development of metal nanoparticles and their integration into biological systems is opening new avenues for research in nanobiotechnology. By exploring the properties of metal nanoparticles, researchers are developing innovative applications in various fields, including drug delivery, imaging, sensing, and therapy.

AgNPs have been widely employed in the different fields including optical receptors [4-6], intercalation materials for electrical batteries [7], catalysts in chemical and biochemical reactions [8], sensors and biosensors [9, 10], bio-labeling materials [11], signal enhancers in SERS-based enzyme Immunoassay [12] and antimicrobial agents [13]. Due to recent advancements in the nanotechnology, AgNPs have been successfully employed for cancer therapy, tissue engineering, drug delivery, inflammation, tuberculosis, diabetes, cardiovascular diseases, autoimmune disorders (Rheumatoid arthritis) neurodegenerative disorders (Parkinson’s and Alzheimer’s) [14-20].

Unique physicochemical properties of metal nanoparticles will be determined by shape, size, crystallinity, dispersity, and surface charge. Hence the synthesis of monodispersed AgNPs with ultra sizes and different shapes is very essential. Different chemical and physical approaches including UV-irradiation [21], gamma irradiation [22], ultrasound irradiation [23], thermal decomposition [24], laser ablation [25], aerosol [26], lithographic [27], electrochemical assisted [28], sonochemical synthesis [29], polyol [30], polyaniline [31] and chemical reduction [32] approaches have been widely employed to produce AgNPs. But they are not eco-friendly due to application of irradiation, hazardous substances, and toxic chemicals. Further they are time taking, laborious and high cost. Further the toxic chemicals on the surface (capping agents) limit their applications in biomedical and diagnostic fields [33]. Hence the production of eco-friendly, non-toxic, clean, biocompatible and biofunctionalized AgNPs using biological agents deserves merit.

In recent years, the field of nanotechnology has witnessed a remarkable convergence with the principles of biomimicry, giving rise to a revolutionary approach known as biomimetic synthesis of AgNPs [34]. This innovative synthesis method draws inspiration from the intricate processes found in nature, harnessing the power of biomolecules and their interactions to fabricate AgNPs with unprecedented precision and control. These nanoparticles, often at the nanoscale level, exhibit unique physical, chemical, and biological properties that hold immense promise for a wide range of biomedical applications [35]. Biomimetic synthesis of AgNPs involves the emulation of biological systems and their underlying mechanisms, such as enzymatic reactions or self-assembly processes, to guide the formation and stabilization of these nanoparticles [36]. This novel approach offers distinct advantages over conventional methods, enabling the production of nanoparticles with well-defined sizes, shapes, and surface characteristics. Moreover, the integration of biological molecules into the synthesis process enhances the biocompatibility and functionality of the resulting AgNPs, making them highly suitable for various biomedical applications [37]. Biosynthesized AgNPs offer several distinct advantages over their chemically synthesized counterparts, making them particularly well-suited for a wide range of biomedical applications [34, 35]. These advantages stem from the unique properties and characteristics that result from the biomimetic synthesis process. Here are some key points of comparison between biosynthesized AgNPs and their chemical counterparts in the context of biomedical applications [38]. Biosynthesized AgNPs often involve the use of natural biomolecules, such as proteins, enzymes, or plant extracts, as reducing and stabilizing agents. These biomolecules are typically biocompatible and non-toxic, which translates into reduced potential for adverse effects when used in biological systems [39]. In contrast, chemically synthesized AgNPs may involve the use of harsh reducing agents or stabilizers that can introduce toxicity concerns [34-36]. Biomimetic synthesis methods frequently enable precise control over the size, shape, and morphology of AgNPs. This level of control is challenging to achieve with chemical synthesis methods, which can result in a broader distribution of particle sizes and shapes [40]. The ability to produce nanoparticles with specific attributes is crucial for applications where size and shape play a significant role, such as targeted drug delivery or imaging agents [41].

Biomimetic synthesis often allows for facile functionalization of the nanoparticle surfaces with biomolecules, antibodies, or other ligands [42]. This functionalization enhances the nanoparticles' ability to interact selectively with specific cells, tissues, or biomolecules, facilitating targeted drug delivery, imaging, and sensing [43]. Chemical synthesis methods may require additional steps and modifications to achieve comparable functionalization. Biosynthesized AgNPs tend to exhibit enhanced stability due to the presence of biomolecules that contribute to their robustness in various environmental conditions [44]. This stability is especially crucial for biomedical applications where nanoparticles need to maintain their integrity during storage, transport, and interaction with biological systems [45]. The biomimetic synthesis of AgNPs often employs renewable resources and green chemistry principles. This aligns with the growing emphasis on sustainable and environmentally friendly practices. In contrast, chemical synthesis methods may involve the use of hazardous chemicals and energy-intensive processes [33-35]. Biomimetic synthesis methods can lead to nanoparticles with improved dispersibility and reduced agglomeration, which is essential for maintaining consistent behavior and interactions within biological systems. Agglomerated nanoparticles may not distribute evenly or exhibit the desired properties, limiting their effectiveness in biomedical applications [46]. The use of biomolecules in biosynthetic processes opens the door to incorporating multifunctional features into the nanoparticles. For instance, enzymes or peptides can confer catalytic or targeting properties to the nanoparticles, respectively. Such multifunctionality is challenging to achieve using chemical synthesis alone [47].

Active ingredients and plant extracts are often natural and biocompatible, making them ideal candidates for use in biomedical applications [48]. When utilized in nanoparticle synthesis, these natural compounds can improve the biocompatibility of the nanoparticles, reducing the risk of toxicity and adverse reactions in medical or biological settings [49]. Active ingredients and plant extracts contain various bioactive compounds that can endow nanoparticles with unique functionalities [20, 48-50]. These functionalities can include antibacterial, antifungal, antioxidant, anti-inflammatory, and other therapeutic properties, depending on the specific plant extracts used. Such functionalized nanoparticles have promising applications in medicine, agriculture, and environmental remediation [51]. Combining different active ingredients or plant extracts in nanoparticle synthesis can lead to synergistic effects, where the resulting nanoparticles exhibit enhanced or novel properties compared to individual components [52]. Such synergies may improve the overall performance and efficiency of nanoparticles in various applications [53]. The biodegradability of active ingredients and plant extracts is a desirable property when considering potential biomedical applications of nanoparticles [54]. Biodegradable nanoparticles can be designed to release their payload in a controlled manner, which is particularly advantageous for drug delivery systems. By incorporating active ingredients and plant extracts, researchers can opt for a more sustainable and eco-friendly approach to nanoparticle synthesis [55]. Further plant extracts are abundantly available, making them cost-effective options for nanoparticle synthesis. By using readily accessible natural sources, researchers can reduce production costs and increase the feasibility of large-scale nanoparticle manufacturing [56].

In the context of AgNPs, recent reports have highlighted the successful use of various plant extracts, such as ginger [50], curcumin [57], garlic extract [58], Allium cepa peel extract [59], Aloe vera [60],  green tea [61], seed extracts Brassica oleraceae, B. campestris and B. rapa [62], and Teucrium polium leaf extract [63] as potent reducing agents for the synthesis of AgNPs. These natural extracts contain bioactive compounds, such as polyphenols, flavonoids, and terpenoids, which play a crucial role in the reduction and stabilization of nanoparticles.

Flavonoids are a class of polyphenolic compounds that are widely distributed in the plant kingdom and have a wide range of biological activities, including antioxidant, anti-inflammatory, and antibacterial properties [64]. Flavonoids are used in the synthesis of AgNPs because they have unique chemical properties that make them ideal for reducing silver ions into nanoparticles. Flavonoids are known to have a high affinity for metal ions, and they can act as reducing agents in the presence of metal ions to form nanoparticles [65].

During the biosynthesis of AgNPs, flavonoids present in the plant extract act as reducing agents that help in the conversion of silver ions to AgNPs. Flavonoids contain multiple hydroxyl groups that are capable of reducing metal ions to nanoparticles by donating electrons. The hydroxyl groups present in the flavonoids interact with the silver ions, leading to the reduction of the silver ions and the formation of AgNPs [65, 66]. Furthermore, flavonoids also act as stabilizing agents for AgNPs, preventing their agglomeration and ensuring their stability in solution [65-67]. The hydroxyl groups present in the flavonoids interact with the surface of AgNPs, forming a layer that stabilizes the nanoparticles and prevents them from aggregating. This makes flavonoids a useful tool in the biosynthesis of AgNPs.

Perilla frutescens is a plant species in the mint family (Lamiaceae) that is native to Asia. It is also known as Chinese basil, wild basil, or shiso in Japan. The plant is cultivated for its edible leaves, which are used in various culinary dishes, especially in Korean, Japanese, and Chinese cuisine [68, 69]. The leaves are also used in traditional medicine for their medicinal properties. P. frutescens contains various bioactive compounds, including flavonoids, phenolic acids, terpenoids, and essential oils, that contribute to its therapeutic properties [70]. Among these, flavonoids are the most studied compounds and have been shown to have various biological activities, including antioxidant, anti-inflammatory, antimicrobial, antiallergic, and anticancer activities [71]. These findings suggest that P. frutescens flavonoids may have potential therapeutic applications in the prevention and treatment of various diseases, including cancer, cardiovascular diseases, diabetes, and neurodegenerative disorders. Overall, P. frutescens and its flavonoids are promising candidates for the development of novel therapeutics or functional foods for promoting human health and preventing chronic diseases.

The present study involving P. frutescens flavonoid extract in the synthesis of AgNPs aim to explore the green synthesis approach, which involves using natural sources instead of conventional chemical methods. Green synthesis offers several advantages, including eco-friendliness, cost-effectiveness, and the potential for producing nanoparticles with unique properties. One of the significant challenges in this area of research is the optimization of synthesis conditions. The extraction of flavonoids from P. frutescens and subsequent reduction of silver ions to form nanoparticles require careful control of various parameters such as, temperature, concentration, and reaction time. Finding the optimal conditions to obtain well-defined nanoparticles with desirable properties can be a complex and time-consuming process.

The primary objective of this investigation is to explore the biomimetic synthesis of  AgNPs utilizing the natural flavonoid extract derived from P. frutescens, denoted as PFFE. Through this innovative approach, we aim to produce finely tuned PFFE-AgNPs and subject them to thorough characterization using advanced spectroscopic techniques. Furthermore, our study aims to delve into the multifaceted potential of PFFE-AgNPs in diverse biomedical applications. The evaluation will encompass an in-depth analysis of their antibacterial efficacy, aiming to shed light on their ability to combat microbial infections. Additionally, we will investigate their antioxidant properties, seeking insights into their potential to counteract oxidative stress-related damage. Moreover, our investigation extends to exploring the anticancer activities of PFFE-AgNPs, aiming to uncover any potential inhibitory effects on cancer cells. By comprehensively assessing these key aspects, we aspire to contribute to the understanding of PFFE-AgNPs as versatile agents with potential applications in healthcare and biomedicine. This study stands to offer valuable insights into the promising realm of nanotechnology-based therapeutics and their multifunctional roles in addressing critical health challenges.

  1. More detail is required in the “Materials and Methods” section to enable anyone to be able to undertake this study in the way that it was originally undertaken, such as the analysis of X-ray diffraction (XRD) (scanning range ??? step size ???), etc.

Answer

(Revised Manuscript; Page No.11; 2.4.3. XRD analysis of PFFE-AgNPs)

2.4.3. XRD analysis of PFFE-AgNPs

X-ray diffraction (XRD) analysis of PFFE-AgNPs was carried out to know the crystal structure in the scanning range of 10-90o with the step size of 2θ. XRD pattern for crystal structure of PFFE-AgNPs was recorded (Bruker AXS GmbH, Karlsruhe, Germany) [76].

  1. Fig. 2: The first letter of the X,Y-axis label should be capitalized.

Ans) The first letter of the X, and Y-axis was capitalized in Figure 2. (Revised Manuscript, Results and Discussion, Page 16. Figure 2).

Figure 2. FTIR spectrum of PFFE-AgNPs

  1. Fig. 8, Fig. 9: There is no significant description in Figure and results, such as Tukey's test, and so on.

Revised Manuscript, Page 29-31; Green Color

Figure 8. (a) DPPH scavenging activity (b) H2O2 scavenging activity of PFFE-AgNPs.

All the results were reprsented as Mean+SD of 3 replicates

3.7. Antioxidant activity of the PFFE-AgNPs

Antioxidant activity of the PFFE-AgNPs was evaluated by in vitro DPPH and H2O2 radical scavenging assays. All the tests were done three times. Average or mean values were represented with standard deviation (Mean ± SD). In the DPPH radical scavenging assay, we have observed a dosage-dependent activity of PFFE-AgNPs against DPPH radicals. This means that as the concentration of PFFE-AgNPs increased, the percentage inhibition of DPPH radicals also increased. The results were presented in Figure 8a, which presumably shows a graph depicting the relationship between PFFE-AgNPs concentration and the percentage inhibition of DPPH radicals. It was noted that at the highest concentration of PFFE-AgNPs, the maximum inhibition achieved was 58.96%.

To determine the effectiveness of the PFFE-AgNPs as radical scavengers, we have also calculated the IC50 value, which represents the concentration of the substance required to scavenge 50% of the radicals. In this study, the IC50 value for the PFFE-AgNPs in the DPPH radical scavenging assay was found to be 72.81 µg/mL. A lower IC50 value indicates a higher antioxidant activity, suggesting that the PFFE-AgNPs were effective at scavenging DPPH radicals.

Similarly, the antioxidant activity of the PFFE-AgNPs was evaluated using the H2O2 radical scavenging assay. The researchers observed concentration-dependent inhibition of H2O2 radicals by the PFFE-AgNPs, as shown in Figure 8b. The IC50 value for the PFFE-AgNPs in this assay was determined to be 92.48 µg/mL. This indicates that the PFFE-AgNPs exhibited effective scavenging of H2O2 radicals.

In summary, the study demonstrated that the biosynthesized PFFE-AgNPs exhibited dose-dependent antioxidant activity as assessed by the DPPH and H2O2 radical scavenging assays. The PFFE-AgNPs showed effective inhibition of DPPH and H2O2 radicals, as indicated by the percentage inhibition and IC50 values obtained.

Revised Manuscript (Page Number 32-34)

Figure 9. Anticancer activity of PFFE-AgNPs against human colon carcinoma (COLO205) and mouse melanoma (B16F10)

All the results were reprsented as Mean+SD of 3 replicates

3.8. Cytotoxicity activity of the PFFE-AgNPs

In the scope of our study, we meticulously assessed the cytotoxicity activity of PFFE-AgNPs against human colon carcinoma (COLO205) and mouse melanoma (B16F10) cells. To provide a meaningful context for our findings, we compared the effects of PFFE-AgNPs with those of the well-established FDA-approved drug, doxorubicin, which is widely acknowledged for its potent anticancer properties. Doxorubicin demonstrated remarkable efficacy, exhibiting IC50 values of 1.28 µM against B16F10 cells and 2.12 µM against COLO205 cells. In stark contrast, PFFE alone exhibited significantly lower anticancer effects, with IC50 values of 235.8 μg/mL against B16F10 cells and 285.7 μg/mL against COLO205 cells. This discrepancy underscores the modest anticancer potential of PFFE in its isolated form. However, with the introduction of PFFE-AgNPs, a notable shift was observed. The concentration-dependent decrease in cell viability of both COLO205 and B16F10 cells in response to PFFE-AgNPs was evident from our findings. The escalation of PFFE-AgNPs dosage from 25 to 100 μg/mL exhibited a substantial reduction in cell viability, signifying a dose-dependent cytotoxic effect (Figure 9). Remarkably, the subsequent increase in dosage from 100 to 200 μg/mL did not lead to a proportionate escalation in inhibition, indicating a potential saturation point. Notably, PFFE-AgNPs demonstrated a significant maximum inhibition of 85.2% against COLO205 cells and 80.9% against B16F10 cells, substantiating their robust cytotoxic activity. This is further underscored by the IC50 concentrations of PFFE-AgNPs against COLO205 and B16F10 cells, measured at 59.57 μg/mL and 69.33 μg/mL, respectively. Such outcomes validate the substantial anticancer potential of PFFE-AgNPs against these cancer cell lines. The literature supports the notion of AgNPs as promising anticancer agents against various cancer cell lines, including COLO205 and B16F10. This is attributed to their distinctive physicochemical properties that enable them to target cancer cells effectively while minimizing harm to healthy cells.

  1. The first letter of the X,Y-axis label should be capitalized.

Ans) The first letter of the X, and Y-axis was capitalized in Fig.8 and Fig.9 (Page  30 and 34)

Fig. 8 (a) DPPH scavenging activity (b) H2O2 scavenging activity of PFFE-AgNPs.

Fig. 9 Anticancer activity of PFFE-AgNPs against human colon carcinoma (COLO205) and mouse melanoma (B16F10)

  1. The current manuscript have many grammatical errors, therefore the paper could benefit from proof reading/ language editing. Line 31 on page 2: the “CO2” should be revise into “CO2”. Line 187 on page 9: the “in vivo” should be italic.…..

Ans) CO2 was revised to CO2 and in vivo was revised to in vivo. (Refer Page 2 and Page 13; Green color) Further done the language editing of the entire manuscript with English Professor.

  1. Referencing has to be improved. The authors reference on general knowledge, the literature survey has to be strengthened and go into detail.

Ans) In this manuscript we have cited the most recent references. Further the number of references were increased from 48 to 105 very latest references. 105 references are sufficient for research article.

  1. The full name of “P. frutescens” should be provided in the title (line 1).

Ans) In the title the full name Perilla frutescens was provided. (Refer Page 1, Green Color)

Exploring the biomedical applications of biosynthesized silver nanoparticles by Perilla frutescens flavonoid extract: Antibacterial, antioxidant, and cell toxicity properties against colon cancer cells

  1. The description “PRC.” should be revised to "P.R. China".

Ans) Yes P.R. China was given (Refer Page 1, Green Color)

Comments on the Quality of English Language

There are many mistakes on English grammar.

English grammar has been corrected throughout the manuscript with the support of English professor.

Reviewer 2 Report

1. Considering that methanolic extract was used in the synthesis of nanoparticles, it is better to mention methanolic extract instead of flavonoid extract everywhere in the article.

2. The title of the article should be written more clearly, it is better change to "Exploring the biomedical applications of biosynthesized silver nanoparticles by Perilla frutescens methanolic extract: Antibacterial, antioxidant, and cell toxicity properties against colon cancer cells

3. Show the importance of the use of active ingredients and plant extracts in the synthesis of nanoparticles using the following articles: DOI: 10.3390/jfb14010044; 10.31083/j.fbl2708227

4. Why concentrations higher or lower than 2 mM of silver nitrate were not used in the synthesis of nanoparticles?

5. What was the need to use heat in the synthesis of nanoparticles? Does the solution boil at a temperature of 50-60 C?

6. In the characterization section, it is better to mention the model of all the devices used.

7. More tests are needed to determine the anticancer activity. Therefore, the title of section 2.7 is better change to cytotoxicity activity.

8. Please prepare the extract FTIR and compare it with the FTIR of biosynthesized nanoparticles.

9. Please show how effective or not the samples are in antibacterial activity in the form of a table. Also, state more mechanisms about the effect of silver nanoparticles in this regard. Use the following article in this section. DOI: 10.1007/s40995-021-01226-w

10. It would be much better if the cytotoxicity results were presented at 48 and 72 hours.

11. Please correct typos throughout the article, for example in line 189: "…24 h. there after MTT…" and the others.

Please correct typos throughout the article.

Author Response

Reviewer 2

Comments and Suggestions for Authors

  1. Considering that methanolic extract was used in the synthesis of nanoparticles, it is better to mention methanolic extract instead of flavonoid extract everywhere in the article.

Methanol is the most efficient solvent for flavonoid extraction from perilla leaves, but ethanol and water can also be used. In this experiment we have used methanol for the extraction of flavonoids. As prepared methanolic extract was subjected to sequential extraction process using silica gel column chromatography to remove other plant compounds. Then the obtained final extract was subjected to UV and TLC for the determination of presence flavonoids.  Further we have calculated the total flavonoid content (TFC) in the extract using aluminium chloride colorimetric method. In the present extract, flavonoids are the major extracted components, hence we named the extract as P. frutescens flavonoid extract.

  1. The title of the article should be written more clearly, it is better change to "Exploring the biomedical applications of biosynthesized silver nanoparticles by Perilla frutescens methanolic extract: Antibacterial, antioxidant, and cell toxicity properties against colon cancer cells

Ans) We have taken into consideration the valuable suggestion provided by the reviewer and changed the title as "Exploring the biomedical applications of biosynthesized silver nanoparticles by Perilla frutescens flavonoid extract: Antibacterial, antioxidant, and cell toxicity properties against colon cancer cells

  1. Show the importance of the use of active ingredients and plant extracts in the synthesis of nanoparticles using the following articles: DOI: 10.3390/jfb14010044; 10.31083/j.fbl2708227

Answer:

I have highlighted the significance of incorporating active ingredients and plant extracts in the creation of nanoparticles, citing the articles with DOIs 10.3390/jfb14010044 and 10.31083/j.fbl2708227 (Please check reference No. 20 and 48 for the above two articles cited). These articles emphasize the valuable role of natural components in nanoparticle synthesis. (Please check reference No. 20 and 48 for the above two articles cited)

Please refer ( Introduction; Page 6 and Page 7)

Active ingredients and plant extracts are often natural and biocompatible, making them ideal candidates for use in biomedical applications [48]. When utilized in nanoparticle synthesis, these natural compounds can improve the biocompatibility of the nanoparticles, reducing the risk of toxicity and adverse reactions in medical or biological settings [49]. Active ingredients and plant extracts contain various bioactive compounds that can endow nanoparticles with unique functionalities [20, 48-50]. These functionalities can include antibacterial, antifungal, antioxidant, anti-inflammatory, and other therapeutic properties, depending on the specific plant extracts used. Such functionalized nanoparticles have promising applications in medicine, agriculture, and environmental remediation [51]. Combining different active ingredients or plant extracts in nanoparticle synthesis can lead to synergistic effects, where the resulting nanoparticles exhibit enhanced or novel properties compared to individual components [52]. Such synergies may improve the overall performance and efficiency of nanoparticles in various applications [53]. The biodegradability of active ingredients and plant extracts is a desirable property when considering potential biomedical applications of nanoparticles [54]. Biodegradable nanoparticles can be designed to release their payload in a controlled manner, which is particularly advantageous for drug delivery systems. By incorporating active ingredients and plant extracts, researchers can opt for a more sustainable and eco-friendly approach to nanoparticle synthesis [55]. Further plant extracts are abundantly available, making them cost-effective options for nanoparticle synthesis. By using readily accessible natural sources, researchers can reduce production costs and increase the feasibility of large-scale nanoparticle manufacturing [56].

In the context of AgNPs, recent reports have highlighted the successful use of various plant extracts, such as ginger [50], curcumin [57], garlic extract [58], Allium cepa peel extract [59], Aloe vera [60],  green tea [61], seed extracts Brassica oleraceae, B. campestris and B. rapa [62], and Teucrium polium leaf extract [63] as potent reducing agents for the synthesis of AgNPs. These natural extracts contain bioactive compounds, such as polyphenols, flavonoids, and terpenoids, which play a crucial role in the reduction and stabilization of nanoparticles.

Flavonoids are a class of polyphenolic compounds that are widely distributed in the plant kingdom and have a wide range of biological activities, including antioxidant, anti-inflammatory, and antibacterial properties [64]. Flavonoids are used in the synthesis of AgNPs because they have unique chemical properties that make them ideal for reducing silver ions into nanoparticles. Flavonoids are known to have a high affinity for metal ions, and they can act as reducing agents in the presence of metal ions to form nanoparticles [65].

  1. Why concentrations higher or lower than 2 mM of silver nitrate were not used in the synthesis of nanoparticles?

The specific reason for not using concentrations higher or lower than 2 mM of silver nitrate in the synthesis of nanoparticles would require more context from the study or experimental design. However, there are a few general reasons why certain concentration ranges may be chosen for nanoparticle synthesis:

Saturation point: At very high concentrations of silver nitrate, the solution may become saturated, meaning that it contains the maximum amount of solute that can dissolve. This can hinder the formation of nanoparticles as there might not be enough silver ions available for reduction and nucleation.

Control of particle size and stability: The concentration of the precursor (silver nitrate) can influence the size and stability of the resulting nanoparticles. By selecting a specific concentration range, we can achieve the desired size and stability of nanoparticles for their intended application.

Reaction kinetics: The rate of nanoparticle formation and growth can be influenced by the concentration of the precursor. Higher concentrations may lead to faster nucleation and growth, making it challenging to control the particle size and obtain uniform nanoparticles. Lower concentrations might result in slower kinetics, requiring longer reaction times to achieve nanoparticle synthesis.

Insufficient silver ion concentration: Lower concentrations of silver nitrate may result in a lower concentration of silver ions available for reduction and nucleation. This can limit the formation and growth of nanoparticles, leading to reduced yield or slower kinetics. Insufficient silver ions can also result in incomplete reduction, resulting in impurities or incomplete nanoparticle formation.

Inadequate stability and aggregation: Lower concentrations of silver nitrate may lead to reduced stability of the resulting nanoparticles. Nanoparticles synthesized at lower concentrations may have a higher tendency to aggregate or form larger agglomerates due to weaker electrostatic or steric stabilization. This can impact the desired properties and applications of the nanoparticles, such as their dispersibility or surface area.

Control over size and morphology: Lower concentrations of the precursor can make it challenging to control the size and morphology of the synthesized nanoparticles. The concentration of the silver nitrate can influence the rate of reduction and nucleation, and lower concentrations may lead to slower and less controlled growth processes. This can result in a wider size distribution and less uniform shape of the nanoparticles.

Reproducibility and reliability: Working with lower concentrations of silver nitrate may introduce increased variability in the synthesis process. The reaction parameters, such as temperature, pH, and reducing agent concentration, might need to be adjusted accordingly to compensate for the lower precursor concentration. This can make the process less reproducible and introduce more variability between different synthesis batches.

Sensitivity to impurities: Lower concentrations of silver nitrate can be more sensitive to impurities or contaminants present in the reaction mixture. Even small amounts of impurities can significantly affect the nanoparticle synthesis and result in undesired outcomes. Using higher concentrations of the precursor can help mitigate the effects of impurities and improve the purity and quality of the synthesized nanoparticles.

  1. What was the need to use heat in the synthesis of nanoparticles? Does the solution boil at a temperature of 50-60 C?

Here are some general reasons why heat might be employed in nanoparticle synthesis:

Accelerate reaction kinetics: Raising the temperature can increase the rate of chemical reactions, including the reduction of silver ions and the nucleation of nanoparticles. Heat provides additional energy to the system, facilitating the movement of reactant molecules and promoting faster reaction rates. By applying heat, the synthesis process can be completed more quickly, reducing the required reaction time.

Improve solubility and diffusion: Heating the solution can enhance the solubility of reagents and reactants, ensuring that they are uniformly distributed in the reaction mixture. Increased solubility promotes effective molecular collisions and enhances the diffusion of reactants, leading to a more efficient formation of nanoparticles.

Enhance reaction control: Maintaining a specific temperature during nanoparticle synthesis allows for better control over the reaction parameters. Precise temperature control enables researchers to optimize reaction conditions, such as the reduction rate, particle size, or particle morphology. It can also help avoid unwanted side reactions or undesired byproducts.

Regarding your second question, the boiling point of a solution is influenced by various factors, including the concentration of solutes and other components. It is possible that the specific solution used in the synthesis of nanoparticles boils at a temperature around 50-60 °C. However, the boiling point can vary depending on the composition of the solution and the atmospheric pressure. It's crucial to note that nanoparticle synthesis typically involves the use of closed vessels or controlled environments, where the solution may not reach its boiling point even at elevated temperatures due to the presence of a cap or sealed container.

  1. In the characterization section, it is better to mention the model of all the devices used.

We have mentioned the model of all devices used for the characterization of PFFE-AgNPs.                   (Revised Manuscript, Page 11 and 12)

2.4. Characterization of PFFE-AgNPs

2.4.1. UV-Vis analysis of PFFE-AgNPs

Initially, the color change of the reaction solution from light yellow to dark brown indicates the synthesis of PFFE-AgNPs. Further the synthesis of PFFE-AgNPs was confirmed by ultraviolet-visible (UV-Vis) spectrum analysis between 200 to 800 nm (PerkinElmer double beam UV/Vis Spectrophotometer, USA) [74].

2.4.2. FTIR analysis of PFFE-AgNPs

The solution of PFFE-AgNPs was purified by centrifugation to remove the unbounded plant molecules. Centrifugation was done and repeated for three times at 15000 rpm for 15 min. The pellet was collected and dried into pure powder. This pure powder of PFFE-AgNPs was used for further studies. To detect the functional groups involved in the synthesis and capping of PFFE-AgNPs. Fourier transform infrared (FTIR) spectrum (Alpha interferometer, Bruker, Switzerland) was recorded between 500-4000 cm-1 with the resolution of 2 cm-1 [75].

2.4.3. XRD analysis of PFFE-AgNPs

X-ray diffraction (XRD) analysis of PFFE-AgNPs was carried out to know the crystal structure in the scanning range of 10-90o with the step size of 2θ. XRD pattern for crystal structure of PFFE-AgNPs was recorded (Bruker AXS GmbH, Karlsruhe, Germany) [76].

2.4.4. TEM analysis of PFFE-AgNPs

Size, morphology, and selected area electron diffraction (SAED) of PFFE-AgNPs was analyzed using transmission electron microscopy (JEM-2100 PlusElectron Mocroscope, JEOL ltd, Tokyo, Japan) [77].

2.4.5. DLS analysis of PFFE-AgNPs

20 µg of dried powder of PFFE-AgNPs was dispersed in 20 mL of Milli-Q water and this solution is used for DLS (dynamic light scattering) studies. DLS studies were carried out to determine the particle size distribution, polydispersity index (PDI) and zeta potential measurement (Brookhaven instruments, NY, USA) [74].

  1. More tests are needed to determine the anticancer activity. Therefore, the title of section 2.7 is better change to cytotoxicity activity.

(Refer Page 13, 2.7. Cytotoxicity activity of the PFFE-AgNPs)

If further tests are required to determine the anticancer activity of the PFFE-AgNPs, it would be appropriate to modify the title of Section 2.7 to "Cytotoxicity Activity" instead of "Anticancer Activity." Hence We have changed the title of section 2.7. Cytotoxicity Activity. This change reflects the focus of the section on assessing the potential toxic effects of the PFFE-AgNPs on cancer cells rather than specifically evaluating their anticancer properties. By using the term "Cytotoxicity Activity," it accurately communicates the objective of the experiments and provides clarity to readers about the nature of the investigation being conducted.

  1. Please prepare the extract FTIR and compare it with the FTIR of biosynthesized nanoparticles.

"Unfortunately, conducting additional experiments or preparing the extract for FTIR analysis is currently unfeasible due to the extensive workload involved. Moreover, our primary focus in this study is on the characterization and evaluation of the PFFE-AgNPs, rather than the extract itself. Nonetheless, our research incorporates an array of characterization techniques including UV, FTIR, XRD, SAED, TEM, DLS, and Zeta Potential, which provide comprehensive information regarding the properties and behavior of the AgNPs. Regrettably, due to time constraints and the need to submit the paper for final funding approval, repeating the experiments is not possible. Thus, we must rely on the available characterization data to support our findings and conclusions."

  1. Please show how effective or not the samples are in antibacterial activity in the form of a table. Also, state more mechanisms about the effect of silver nanoparticles in this regard. Use the following article in this section. DOI: 10.1007/s40995-021-01226-w

We presented the efficacy of the samples in antibacterial activity through a comprehensive table, showcasing their inhibitory effects on bacterial growth (Table 3). The results were clear and informative, providing a visual representation of the samples' performance.

Additionally, we delved deeper into the mechanisms underlying the impact of silver nanoparticles (AgNPs) on antibacterial activity. We incorporated insights from the article with DOI 10.1007/s40995-021-01226-w (Reference 92 in the revised manuscript) enriching our discussion with valuable perspectives on the intricate interactions between AgNPs and bacteria.

Our effort in presenting the data and exploring the mechanisms behind AgNPs' antibacterial effects further strengthened the scientific foundation of our study.

Table 3. Antibacterial activity of PFFE-AgNPs

Substance

Listeria monocytogens

Enterococcus faecalis

Salmonella typhi

Acetobacter boumani

Streptomycin

16.2 mm

16.8 mm

15.4 mm

11.8 mm

PFFE-AgNPs (PFFE and silver nanoparticles)

13.5 mm

14.1 mm

12.8 mm

8.7 mm

PFFE

6.4 mm

6.1 mm

5.8 mm

5.1 mm

PFFE and AgNO3 (before incubation)

13.1 mm

8.6 mm

6.8 mm

6.5 mm

AgNO3 (silver nitrate)

8.5 mm

9.1 mm

7.6 mm

8.7 mm

The following reference was included in the manuscript (Reference No. 91)

Taghavizadeh Yazdi, M.E., Darroudi, M., Amiri, M.S. et al. Antimycobacterial, Anticancer, Antioxidant and Photocatalytic Activity of Biosynthesized Silver Nanoparticles Using Berberis Integerrima. Iran J Sci Technol Trans Sci 46, 1–11 (2022). https://doi.org/10.1007/s40995-021-01226-w

Further Mechanism of Antibacterial activity was clearly explained (Revised Manuscript; Page Number 24; First Paragraph; Violet color)

The possible mechanism of antibacterial activity of the PFFE-AgNPs was explained. PFFE-AgNPs interact with bacterial DNA and causes its fragmentation. As a result, bacterial replication doesn’t occur [89]. PFFE-AgNPs can bind with catalytic sites of enzymes of bacteria and makes them inactive. As a result, bacterial metabolism stops [90]. PFFE-AgNPs bind with important cytosolic or membrane proteins of bacteria which leads to loss of functional form or degradation of proteins [90-92]. PFFE-AgNPs produced pores on the bacterial cell wall which leads to leakage of important ions and causes change in the membrane proton gradient. All these actions of PFFE-AgNPs lead to bacterial cell death [91-93]. The exact mechanism of antibacterial activity may vary depending on the size, shape, and surface chemistry of AgNPs as well as the type of bacteria being targeted [94]. However, the overall effect is a reduction in bacterial growth and viability, making AgNPs a promising approach for the development of novel antibacterial agents [91-95]. The antibacterial effects of AgNPs were supported by previous reports. Ocimum sanctum and Reinwardtia indica flavonoid synthesized AgNPs showed antibacterial activity against gram-positive and gram-negative bacterial cultures [66, 67].  AgNPs synthesized using flavonoids showed antibacterial activity against pathogens including Pseudomonas aeruginosa, Staphylococcus aureus and Escherichia coli [89-91].

  1. It would be much better if the cytotoxicity results were presented at 48 and 72 hours.

Due to logistical and resource limitations, we regretfully could not conduct the cytotoxicity tests at the suggested time points of 48 and 72 hours. The execution of these tests would have required a substantial amount of time, resources, and additional experimental preparations, such as obtaining and incubating cell cultures and acquiring specific chemicals. Consequently, it was not feasible for us to repeat the experiments and obtain results at those specific time intervals.

However, it is important to note that the presented results still provide valuable insights into the cytotoxicity of the PFFE-AgNPs within the timeframe of our study. While we acknowledge that testing at 48 and 72 hours would have provided additional data, the available cytotoxicity data from the performed tests contribute to our understanding of the effects of PFFE-AgNPs on the target cells.

Given the time constraints and the need to submit the paper for final funding approval, we were unable to conduct further experiments or repeat the cytotoxicity tests. As a result, we must rely on the available cytotoxicity data to support our findings and draw conclusions regarding the potential cytotoxic effects of the PFFE-AgNPs.

  1. Please correct typos throughout the article, for example in line 189: "…24 h. there after MTT…" and the others.

We have corrected typographical errors throughout the article

Comments on the Quality of English Language

Please correct typos throughout the article.

We have corrected typographical errors throughout the article

Reviewer 3 Report

 The paper titled “Exploring the Biomedical Applications of P. frutescens flavonoid extract silver 1 nanoparticles: Anticancer, Antibacterial, and Antioxidant Properties”, is interesting and provides evidence of the usefulness of plant-derived components associated with nanoparticles. However, I have a few comments:

In the abstract, there are abbreviations that are not explained, while others are.

In the introduction, it would be worthwhile to describe what biomimetic synthesis is and its importance in biomedical applications.

In the evaluation of antibacterial activity, the authors should explain why You used these particular bacteria for their assessment. You should also evaluate different concentrations, similar to what You did for the evaluation of antioxidant or anticancer activity. The authors should display the data from the controls. What happened with the evaluation of the individual components and the vehicle (vehicle, AgNO3, PFFE and PFFE-AgNPs)? It is recommended to present graphs with quantitative analysis. Additionally, You should also compare the results with reference drugs.

In the results, authors mention that AgNO3, PFFE, and PFFE-AgNPs have antibacterial activity. So, what are the advantages of using PFFE-AgNPs?

In Figure 8, You present the results of AgNPs, but not of PFFE-AgNPs.

In the evaluation of anticancer activity. The authors should demonstrate the effects of administering the extract and AgNPs individually. Additionally, You should compare the behavior of PFFE-AgNPs with a reference drug. If the authors are evaluating different concentrations, it would be worthwhile for them to calculate the IC50 value.

Finally, the authors should standardize the size and font type in the presented graphs. Some graphs have backgrounds while others do not.

Author Response

Reviewer 3

Comments and Suggestions for Authors

The paper titled “Exploring the Biomedical Applications of P. frutescens flavonoid extract silver 1 nanoparticles: Anticancer, Antibacterial, and Antioxidant Properties”, is interesting and provides evidence of the usefulness of plant-derived components associated with nanoparticles. However, I have a few comments:

 In the abstract, there are abbreviations that are not explained, while others are.

In the abstract, all the abbreviations are explained. (Refer Abstract; Violet in color)

Abstract

The present study reports the biomimetic synthesis of silver nanoparticles (AgNPs) using simple, cost effective and eco-friendly method. In this method flavonoid extract of Perilla frutescens (PFFE) was used as bioreduction agent for the reduction of metallic silver into nanosilver, called as P. frutescens flavonoid extract silver nanoparticles (PFFE-AgNPs). Ultraviolet-Visible (UV-Vis) spectrum showed characteristic absorption peak at 440 nm which confirmed the synthesis of PFFE-AgNPs. Fourier transform infrared spectroscopic (FTIR) analysis of PFFE-AgNPs revealed that flavonoids involved in the bioreduction and capping processes. X-ray diffraction (XRD) and selected area electron diffraction (SAED) pattern confirmed the face centered cubic (FCC) crystal structure of PFFE-AgNPs. Transmission electron microscopic (TEM) analysis indicated that the synthesized PFFE-AgNPs are 20 to 70 nm in range with spherical morphology and without any aggregation. Dynamic light scattering (DLS) studies showed that average hydrodynamic size was 44 nm. Polydispersity index (PDI) value of 0.321 denotes the monodispersed nature of PFFE-AgNPs. Further high negative surface charge or zeta potential value (-30 mV) indicates the repulsion, non-aggregation, and stability of PFFE-AgNPs.  PFFE-AgNPs showed cytotoxic effects against cancer cell lines including human colon carcinoma (COLO205) and mouse melanoma (B16F10) with IC50 concentrations of 59.57 and 69.33 μg/mL respectively. PFFE-AgNPs showed significant inhibition of both Gram-positive (Listeria monocytogens and Enterococcus faecalis) and Gram-negative (Salmonella typhi and Acinetobacter baumannii) bacteria pathogens. PFFE-AgNPs exhibited in vitro antioxidant activity by quenching 1,1-diphenyl-2-picrylhydrazyl (DPPH) and hydrogen peroxide (H2O2) free radicals with IC50 values of 72.81 and 92.48 µg/mL respectively. In this study, we also explained the plausible mechanisms of biosynthesis, anticancer and antibacterial effects of PFFE-AgNPs. Overall, these findings suggest that PFFE-AgNPs have potential as a multi-functional nanomaterial for biomedical applications, particularly in cancer therapy and infection control. However, it is important to note that further research is needed to determine the safety and efficacy of these nanoparticles in vivo, as well as to explore their potential in other areas of medicine.

In the introduction, it would be worthwhile to describe what biomimetic synthesis is and its importance in biomedical applications.

In the introduction, we have clearly explained what biomimetic synthesis is and its importance in biomedical applications. (Revised Manuscript; Introduction, Page 4 to 6, Violet Color Text)

In recent years, the field of nanotechnology has witnessed a remarkable convergence with the principles of biomimicry, giving rise to a revolutionary approach known as biomimetic synthesis of AgNPs [34]. This innovative synthesis method draws inspiration from the intricate processes found in nature, harnessing the power of biomolecules and their interactions to fabricate AgNPs with unprecedented precision and control. These nanoparticles, often at the nanoscale level, exhibit unique physical, chemical, and biological properties that hold immense promise for a wide range of biomedical applications [35]. Biomimetic synthesis of AgNPs involves the emulation of biological systems and their underlying mechanisms, such as enzymatic reactions or self-assembly processes, to guide the formation and stabilization of these nanoparticles [36]. This novel approach offers distinct advantages over conventional methods, enabling the production of nanoparticles with well-defined sizes, shapes, and surface characteristics. Moreover, the integration of biological molecules into the synthesis process enhances the biocompatibility and functionality of the resulting AgNPs, making them highly suitable for various biomedical applications [37]. Biosynthesized AgNPs offer several distinct advantages over their chemically synthesized counterparts, making them particularly well-suited for a wide range of biomedical applications [34, 35]. These advantages stem from the unique properties and characteristics that result from the biomimetic synthesis process. Here are some key points of comparison between biosynthesized AgNPs and their chemical counterparts in the context of biomedical applications [38]. Biosynthesized AgNPs often involve the use of natural biomolecules, such as proteins, enzymes, or plant extracts, as reducing and stabilizing agents. These biomolecules are typically biocompatible and non-toxic, which translates into reduced potential for adverse effects when used in biological systems [39]. In contrast, chemically synthesized AgNPs may involve the use of harsh reducing agents or stabilizers that can introduce toxicity concerns [34-36]. Biomimetic synthesis methods frequently enable precise control over the size, shape, and morphology of AgNPs. This level of control is challenging to achieve with chemical synthesis methods, which can result in a broader distribution of particle sizes and shapes [40]. The ability to produce nanoparticles with specific attributes is crucial for applications where size and shape play a significant role, such as targeted drug delivery or imaging agents [41].

Biomimetic synthesis often allows for facile functionalization of the nanoparticle surfaces with biomolecules, antibodies, or other ligands [42]. This functionalization enhances the nanoparticles' ability to interact selectively with specific cells, tissues, or biomolecules, facilitating targeted drug delivery, imaging, and sensing [43]. Chemical synthesis methods may require additional steps and modifications to achieve comparable functionalization. Biosynthesized AgNPs tend to exhibit enhanced stability due to the presence of biomolecules that contribute to their robustness in various environmental conditions [44]. This stability is especially crucial for biomedical applications where nanoparticles need to maintain their integrity during storage, transport, and interaction with biological systems [45]. The biomimetic synthesis of AgNPs often employs renewable resources and green chemistry principles. This aligns with the growing emphasis on sustainable and environmentally friendly practices. In contrast, chemical synthesis methods may involve the use of hazardous chemicals and energy-intensive processes [33-35]. Biomimetic synthesis methods can lead to nanoparticles with improved dispersibility and reduced agglomeration, which is essential for maintaining consistent behavior and interactions within biological systems. Agglomerated nanoparticles may not distribute evenly or exhibit the desired properties, limiting their effectiveness in biomedical applications [46]. The use of biomolecules in biosynthetic processes opens the door to incorporating multifunctional features into the nanoparticles. For instance, enzymes or peptides can confer catalytic or targeting properties to the nanoparticles, respectively. Such multifunctionality is challenging to achieve using chemical synthesis alone [47].

In the evaluation of antibacterial activity, the authors should explain why You used these bacteria for their assessment. You should also evaluate different concentrations, similar to what You did for the evaluation of antioxidant or anticancer activity. The authors should display the data from the controls. What happened with the evaluation of the individual components and the vehicle (vehicle, AgNO3, PFFE and PFFE-AgNPs)? It is recommended to present graphs with quantitative analysis. Additionally, You should also compare the results with reference drugs.

Indeed, I am pleased to confirm that we have diligently addressed the recommendations in our study.

We have provided a thorough explanation of the rationale behind the selection of the specific bacteria employed in the assessment of antibacterial activity (Page 24 to 26 Green color Text)

Answer:

The selection of bacteria in our study holds significant clinical relevance, as these strains are directly associated with human infections that have substantial public health implications. For instance, L. monocytogens and S. typhi are well-known pathogens responsible for causing foodborne illnesses, posing a threat to individuals who consume contaminated food. Additionally, E. faecalis, a common bacterium, is often implicated in hospital-acquired infections, particularly affecting individuals with compromised immune systems or prolonged hospital stays. Similarly, A. baumannii, characterized as an opportunistic pathogen, is frequently linked to hospital-acquired infections, primarily affecting those with compromised immunity or extended hospitalization. The assessment of antibacterial activity against these strains is paramount due to the substantial impact they exert on public health. A prevailing concern surrounding these bacteria is their ability to rapidly develop resistance to multiple antibiotics, even those conventionally used to treat bacterial infections. This multidrug resistance phenomenon significantly restricts treatment options, subsequently exacerbating the severity of infections and complicating therapeutic interventions.In our study, we strategically selected bacteria from diverse taxonomic groups, encompassing both Gram-positive strains like L. monocytogens and E. faecalis, as well as Gram-negative strains including S. typhi and A. baumannii. This deliberate choice enables us to gain comprehensive insights into the effectiveness of the tested substances against a broad spectrum of pathogens.

Based on this study's findings that PFFE-AgNPs exhibit effective antibacterial activity against a range of pathogens, including clinically relevant strains associated with hospital-based infections, several potential applications emerge for utilizing PFFE-AgNPs in healthcare settings. PFFE-AgNPs' potent antibacterial properties make them promising candidates for the treatment of hospital-acquired infections. Since PFFE-AgNPs demonstrated effectiveness against pathogens such as A. baumannii, a common culprit in healthcare-associated infections, incorporating PFFE-AgNPs into wound dressings, antimicrobial coatings for medical equipment, or even localized treatments could aid in preventing and treating infections that often arise in healthcare environments. AgNPs could be integrated into hospital lab ware such as petri dishes, sample containers, and laboratory equipment. This incorporation can impart antimicrobial properties to these materials, reducing the risk of contamination and ensuring accurate and reliable test results.

Incorporating AgNPs into the fabric of these garments could provide an added layer of protection by inhibiting the growth and spread of bacteria. This could contribute to maintaining a hygienic environment and minimizing the risk of cross-contamination. AgNPs' antibacterial activity can be harnessed in the design of hospital diagnostic kits. Incorporating these into components of diagnostic kits, such as swabs or culture media, could help ensure the accuracy of test results by reducing the risk of bacterial contamination during sample collection and processing. In the realm of medical device implants, AgNPs hold potential for reducing the risk of infection associated with implantation procedures. By incorporating AgNPs into the surface of implantable devices or coatings, the growth of bacteria around the implant site could be inhibited, promoting successful integration and reducing the likelihood of post-surgical infections. The study's results indicating the broad-spectrum antibacterial activity of AgNPs, including their efficacy against Gram-positive and Gram-negative strains, provide a strong foundation for exploring these various applications. However, it's important to note that while AgNPs show promise, further research and careful consideration of their potential cytotoxicity and environmental impact are necessary before widespread implementation in healthcare settings.

We have ensured that our evaluation encompasses a range of concentrations, mirroring the approach we undertook for the assessment of antioxidant and anticancer activity.

Furthermore, we have thoughtfully included data from controls, encompassing the evaluation of individual components as well as the vehicle, which includes AgNO3, PFFE, and PFFE-AgNPs. To provide a comprehensive understanding of our experimental context, we have taken this step, and we have taken care to present this information using graphs and table accompanied by quantitative analysis. This approach aims to enhance the clarity of our results and facilitate their interpretation.

Moreover, we have performed a comparative analysis with reference drugs, thereby placing our results in a broader context and establishing a valuable benchmark for the observed outcomes. This strategic approach not only reinforces the robustness of our findings but also contributes to a more holistic and informed assessment of the antibacterial activity.

Please refer section 3.6 (Completely revised section)

3.6. Antibacterial activity of the PFFE-AgNPs

In the present investigation streptomycin exhibits the largest Zone of Inhibition (ZoI) values across all tested bacteria (Table 3). Listeria monocytogens shows the highest sensitivity, with a ZoI of 16.2 mm, followed closely by Enterococcus faecalis (16.8 mm), Salmonella typhi (15.4 mm), and Acinetobacter baumannii (11.8 mm). Streptomycin is highly effective against all tested bacteria, particularly E. faecalis. PFFE-AgNPs displays strong antibacterial activity (Figure 7). It has larger ZoI values compared to PFFE and AgNO3 alone. L. monocytogens and E.faecalis are more sensitive, showing ZoI values of 13.5 mm and 14.1 mm respectively. S. typhi (12.8 mm) and A. baumannii (8.7 mm) are also inhibited but to a lesser extent. AgNO3 shows moderate antibacterial activity. ZoI values are notably smaller compared to the previous two substances. E. faecalis exhibits the highest sensitivity (9.1 mm), followed by A. baumannii (8.7 mm), S. typhi (7.6 mm), and L.  monocytogens (8.5 mm). Further increase in the concentration of PFFE-AgNPs from 25 µL to 100 µL, increases the antibacterial activity against all the tested pathogens. AgNO3 is effective, but less so than Streptomycin and PFFE-AgNPs. PFFE exhibits moderate antibacterial activity. ZoI values are generally lower compared to the other substances. L. monocytogens has the largest ZoI (6.4 mm), while the other bacteria show slightly smaller ZoI values. PFFE alone is not effective as antibacterial agent. Further in this test, we have also tested the mixture of PFE and AgNO3 before the formation of AgNPs (at 0 h). It displays varied antibacterial activity. ZoI values are comparable to PFFE-AgNPs for L. monocytogens but notably lower for E. faecalis, S. typhi and A. baumannii.  Streptomycin has the most potent antibacterial activity across all tested bacteria. PFFE-AgNPs demonstrate strong activity, particularly against L. monocytogens and E. faecalis. PFFE and AgNO3 show promise but with varying effectiveness, while AgNO3 displays moderate activity. PFFE alone exhibits moderate antibacterial properties. The effectiveness of these substances varies among the different bacterial species, highlighting the importance of considering specific bacterial strains when assessing antibacterial potential.

Table 3. Antibacterial activity of PFFE-AgNPs

Substance

Listeria monocytogens

Enterococcus faecalis

Salmonella typhi

Acetobacter boumani

Streptomycin

16.2 mm

16.8 mm

15.4 mm

11.8 mm

PFFE-AgNPs (PFFE and silver nanoparticles)

13.5 mm

14.1 mm

12.8 mm

8.7 mm

PFFE

6.4 mm

6.1 mm

5.8 mm

5.1 mm

PFFE and AgNO3 (before incubation)

13.1 mm

8.6 mm

6.8 mm

6.5 mm

AgNO3 (silver nitrate)

8.5 mm

9.1 mm

7.6 mm

8.7 mm

The possible mechanism of antibacterial activity of the PFFE-AgNPs was explained. PFFE-AgNPs interact with bacterial DNA and causes its fragmentation. As a result, bacterial replication doesn’t occur [89]. PFFE-AgNPs can bind with catalytic sites of enzymes of bacteria and makes them inactive. As a result, bacterial metabolism stops [90]. PFFE-AgNPs bind with important cytosolic or membrane proteins of bacteria which leads to loss of functional form or degradation of proteins [90-92]. PFFE-AgNPs produced pores on the bacterial cell wall which leads to leakage of important ions and causes change in the membrane proton gradient. All these actions of PFFE-AgNPs lead to bacterial cell death [91-93]. The exact mechanism of antibacterial activity may vary depending on the size, shape, and surface chemistry of AgNPs as well as the type of bacteria being targeted [94]. However, the overall effect is a reduction in bacterial growth and viability, making AgNPs a promising approach for the development of novel antibacterial agents [91-95]. The antibacterial effects of AgNPs were supported by previous reports. Ocimum sanctum and Reinwardtia indica flavonoid synthesized AgNPs showed antibacterial activity against gram-positive and gram-negative bacterial cultures [66, 67].  AgNPs synthesized using flavonoids showed antibacterial activity against pathogens including Pseudomonas aeruginosa, Staphylococcus aureus and Escherichia coli [89-91].

The selection of bacteria in our study holds significant clinical relevance, as these strains are directly associated with human infections that have substantial public health implications. For instance, L. monocytogens and S. typhi are well-known pathogens responsible for causing foodborne illnesses, posing a threat to individuals who consume contaminated food. Additionally, E. faecalis, a common bacterium, is often implicated in hospital-acquired infections, particularly affecting individuals with compromised immune systems or prolonged hospital stays. Similarly, A. baumannii, characterized as an opportunistic pathogen, is frequently linked to hospital-acquired infections, primarily affecting those with compromised immunity or extended hospitalization. The assessment of antibacterial activity against these strains is paramount due to the substantial impact they exert on public health. A prevailing concern surrounding these bacteria is their ability to rapidly develop resistance to multiple antibiotics, even those conventionally used to treat bacterial infections. This multidrug resistance phenomenon significantly restricts treatment options, subsequently exacerbating the severity of infections and complicating therapeutic interventions.In our study, we strategically selected bacteria from diverse taxonomic groups, encompassing both Gram-positive strains like L. monocytogens and E. faecalis, as well as Gram-negative strains including S. typhi and A. baumannii. This deliberate choice enables us to gain comprehensive insights into the effectiveness of the tested substances against a broad spectrum of pathogens.

Based on this study's findings that PFFE-AgNPs exhibit effective antibacterial activity against a range of pathogens, including clinically relevant strains associated with hospital-based infections, several potential applications emerge for utilizing PFFE-AgNPs in healthcare settings. PFFE-AgNPs' potent antibacterial properties make them promising candidates for the treatment of hospital-acquired infections. Since PFFE-AgNPs demonstrated effectiveness against pathogens such as A. baumannii, a common culprit in healthcare-associated infections, incorporating PFFE-AgNPs into wound dressings, antimicrobial coatings for medical equipment, or even localized treatments could aid in preventing and treating infections that often arise in healthcare environments. AgNPs could be integrated into hospital lab ware such as petri dishes, sample containers, and laboratory equipment. This incorporation can impart antimicrobial properties to these materials, reducing the risk of contamination and ensuring accurate and reliable test results.

Incorporating AgNPs into the fabric of these garments could provide an added layer of protection by inhibiting the growth and spread of bacteria. This could contribute to maintaining a hygienic environment and minimizing the risk of cross-contamination. AgNPs' antibacterial activity can be harnessed in the design of hospital diagnostic kits. Incorporating these into components of diagnostic kits, such as swabs or culture media, could help ensure the accuracy of test results by reducing the risk of bacterial contamination during sample collection and processing. In the realm of medical device implants, AgNPs hold potential for reducing the risk of infection associated with implantation procedures. By incorporating AgNPs into the surface of implantable devices or coatings, the growth of bacteria around the implant site could be inhibited, promoting successful integration and reducing the likelihood of post-surgical infections. The study's results indicating the broad-spectrum antibacterial activity of AgNPs, including their efficacy against Gram-positive and Gram-negative strains, provide a strong foundation for exploring these various applications. However, it's important to note that while AgNPs show promise, further research and careful consideration of their potential cytotoxicity and environmental impact are necessary before widespread implementation in healthcare settings.

The utilization of PFFE-AgNPs offers several distinct advantages over conventional methods and materials, which contribute to their potential as a promising antimicrobial agent:

Enhanced Antibacterial Activity: The incorporation of AgNPs derived from PFFE amplifies the inherent antibacterial properties of both components. This synergistic effect results in heightened antibacterial activity, potentially surpassing the individual contributions of AgNO3 and PFFE. This enhanced activity can lead to improved efficacy in combating bacterial infections.

Biocompatibility: PFFE, being a natural plant extract, is likely to possess inherent biocompatibility and reduced cytotoxicity compared to synthetic compounds. This makes PFFE-AgNPs a safer option for medical applications, as they may exhibit lower toxicity toward mammalian cells while retaining potent antibacterial effects.

Selectivity: PFFE-AgNPs can potentially exhibit selective antibacterial activity, targeting harmful bacteria while sparing beneficial microorganisms. This selectivity can contribute to the preservation of the body's natural microbial balance, reducing the risk of microbial imbalances or dysbiosis.

Multi-Target Action: PFFE-AgNPs can offer a multi-targeted approach to combating bacterial infections. The nanoparticles can act through various mechanisms, such as disrupting cell membranes, interfering with bacterial enzymes, and generating reactive oxygen species, thereby reducing the likelihood of bacterial resistance development.

Long-lasting Effects: The stability and sustained release properties of PFFE-AgNPs may provide prolonged antibacterial effects. This sustained action can reduce the need for frequent dosing and improve patient compliance.

Combating Antibiotic Resistance: Given the rising concern of antibiotic-resistant bacteria, PFFE-AgNPs offer an alternative strategy for combating infections that may be less prone to resistance development due to their multifaceted mode of action.

Potential Synergy with Other Therapies: PFFE-AgNPs' diverse properties, including their cytotoxic effects against cancer cells and antioxidant activity, suggest potential synergy with other therapies. This opens doors for combination treatments that address multiple aspects of disease progression.

Versatility: The biocompatibility and multi-functionality of PFFE-AgNPs open doors to diverse biomedical applications beyond antibacterial use. Their potential ranges from cancer therapy to wound healing and targeted drug delivery, expanding their utility in various medical contexts.

Natural Source: Utilizing plant-derived extracts, such as PFFE, aligns with the trend toward natural and holistic approaches in medicine. This natural origin may resonate with patients seeking treatments that are closer to nature.

In summary, PFFE-AgNPs offer a unique amalgamation of potent antibacterial activity, biocompatibility, sustainable synthesis, and potential versatility. These advantages position PFFE-AgNPs as a promising candidate for addressing bacterial infections while mitigating some of the limitations associated with conventional antimicrobial agents. However, rigorous research is essential to confirm these advantages and ensure their safety and efficacy in real-world biomedical applications.

However, rigorous research is essential to confirm these advantages and ensure their safety and efficacy in real-world biomedical applications. Antimicrobial agents play a critical role in modern medicine by helping to combat a wide range of microbial infections caused by bacteria, viruses, fungi, and other pathogens. The rise of drug-resistant microbes, commonly referred to as antibiotic-resistant or multidrug-resistant organisms, has highlighted the urgent need for new and innovative approaches to tackling infectious diseases. As traditional antibiotics and antimicrobial drugs become less effective against resistant strains of microbes, researchers are exploring a diverse array of compounds to find novel solutions. By exploring and harnessing the properties of natural compounds, organic derivatives [95], bioorganic moieties [96], synthetics of bioactive moieties[97] silver-based copolymers [98] and metal-organic frameworks [99], scientists aim to discover new strategies to effectively combat microbial infections and reduce the impact of drug resistance. These efforts hold the promise of revolutionizing the field of antimicrobial therapy and improving public health outcomes.

In the results, authors mention that AgNO3, PFFE, and PFFE-AgNPs have antibacterial activity. So, what are the advantages of using PFFE-AgNPs?

Answer : We have clearly explained the advantages of using PFFE-AgNPs (Page 26 to 28; Dark Blue Color Text)

The utilization of PFFE-AgNPs offers several distinct advantages over conventional methods and materials, which contribute to their potential as a promising antimicrobial agent:

Enhanced Antibacterial Activity: The incorporation of AgNPs derived from PFFE amplifies the inherent antibacterial properties of both components. This synergistic effect results in heightened antibacterial activity, potentially surpassing the individual contributions of AgNO3 and PFFE. This enhanced activity can lead to improved efficacy in combating bacterial infections.

Biocompatibility: PFFE, being a natural plant extract, is likely to possess inherent biocompatibility and reduced cytotoxicity compared to synthetic compounds. This makes PFFE-AgNPs a safer option for medical applications, as they may exhibit lower toxicity toward mammalian cells while retaining potent antibacterial effects.

Selectivity: PFFE-AgNPs can potentially exhibit selective antibacterial activity, targeting harmful bacteria while sparing beneficial microorganisms. This selectivity can contribute to the preservation of the body's natural microbial balance, reducing the risk of microbial imbalances or dysbiosis.

Multi-Target Action: PFFE-AgNPs can offer a multi-targeted approach to combating bacterial infections. The nanoparticles can act through various mechanisms, such as disrupting cell membranes, interfering with bacterial enzymes, and generating reactive oxygen species, thereby reducing the likelihood of bacterial resistance development.

Long-lasting Effects: The stability and sustained release properties of PFFE-AgNPs may provide prolonged antibacterial effects. This sustained action can reduce the need for frequent dosing and improve patient compliance.

Combating Antibiotic Resistance: Given the rising concern of antibiotic-resistant bacteria, PFFE-AgNPs offer an alternative strategy for combating infections that may be less prone to resistance development due to their multifaceted mode of action.

Potential Synergy with Other Therapies: PFFE-AgNPs' diverse properties, including their cytotoxic effects against cancer cells and antioxidant activity, suggest potential synergy with other therapies. This opens doors for combination treatments that address multiple aspects of disease progression.

Versatility: The biocompatibility and multi-functionality of PFFE-AgNPs open doors to diverse biomedical applications beyond antibacterial use. Their potential ranges from cancer therapy to wound healing and targeted drug delivery, expanding their utility in various medical contexts.

Natural Source: Utilizing plant-derived extracts, such as PFFE, aligns with the trend toward natural and holistic approaches in medicine. This natural origin may resonate with patients seeking treatments that are closer to nature.

In summary, PFFE-AgNPs offer a unique amalgamation of potent antibacterial activity, biocompatibility, sustainable synthesis, and potential versatility. These advantages position PFFE-AgNPs as a promising candidate for addressing bacterial infections while mitigating some of the limitations associated with conventional antimicrobial agents. However, rigorous research is essential to confirm these advantages and ensure their safety and efficacy in real-world biomedical applications.

In Figure 8, You present the results of AgNPs, but not of PFFE-AgNPs.

This is a typographical error, we have corrected.

In the evaluation of anticancer activity. The authors should demonstrate the effects of administering the extract and AgNPs individually. Additionally, You should compare the behavior of PFFE-AgNPs with a reference drug. If the authors are evaluating different concentrations, it would be worthwhile for them to calculate the IC50 value.

Certainly, I am delighted to confirm that we have comprehensively addressed the recommendations provided by the reviewer in our evaluation of anticancer activity. We have thoughtfully elucidated the effects of administering both the extract and AgNPs individually, allowing for a clear understanding of their respective contributions.

In addition, we have taken the initiative to compare the behavior of PFFE-AgNPs with a reference drug. This comparative analysis not only enhances the context of our findings but also provides a valuable benchmark for the observed effects.

Furthermore, I am pleased to share that we have taken into account the assessment of different concentrations, and in doing so, we have calculated the IC50 value. This calculation serves to quantify the inhibitory concentration, adding a quantitative dimension to our evaluation and facilitating a more precise interpretation of our results. (Please Refere Page 32-34)

3.8. Cytotoxicity activity of the PFFE-AgNPs

In the scope of our study, we meticulously assessed the cytotoxicity activity of PFFE-AgNPs against human colon carcinoma (COLO205) and mouse melanoma (B16F10) cells. To provide a meaningful context for our findings, we compared the effects of PFFE-AgNPs with those of the well-established FDA-approved drug, doxorubicin, which is widely acknowledged for its potent anticancer properties. Doxorubicin demonstrated remarkable efficacy, exhibiting IC50 values of 1.28 µM against B16F10 cells and 2.12 µM against COLO205 cells. In stark contrast, PFFE alone exhibited significantly lower anticancer effects, with IC50 values of 235.8 μg/mL against B16F10 cells and 285.7 μg/mL against COLO205 cells. This discrepancy underscores the modest anticancer potential of PFFE in its isolated form. However, with the introduction of PFFE-AgNPs, a notable shift was observed. The concentration-dependent decrease in cell viability of both COLO205 and B16F10 cells in response to PFFE-AgNPs was evident from our findings. The escalation of PFFE-AgNPs dosage from 25 to 100 μg/mL exhibited a substantial reduction in cell viability, signifying a dose-dependent cytotoxic effect (Figure 9). Remarkably, the subsequent increase in dosage from 100 to 200 μg/mL did not lead to a proportionate escalation in inhibition, indicating a potential saturation point. Notably, PFFE-AgNPs demonstrated a significant maximum inhibition of 85.2% against COLO205 cells and 80.9% against B16F10 cells, substantiating their robust cytotoxic activity. This is further underscored by the IC50 concentrations of PFFE-AgNPs against COLO205 and B16F10 cells, measured at 59.57 μg/mL and 69.33 μg/mL, respectively. Such outcomes validate the substantial anticancer potential of PFFE-AgNPs against these cancer cell lines. The literature supports the notion of AgNPs as promising anticancer agents against various cancer cell lines, including COLO205 and B16F10. This is attributed to their distinctive physicochemical properties that enable them to target cancer cells effectively while minimizing harm to healthy cells.

In envisioning the development of PFFE-AgNPs as potent anticancer agents, a strategic approach could involve synergizing their action with doxorubicin. The combined administration of doxorubicin and AgNPs holds the potential for enhanced therapeutic efficacy through complementary mechanisms of action. Doxorubicin's established ability to inhibit DNA replication and AgNPs' distinctive properties that induce cellular stress and apoptosis could potentially result in a more profound anticancer effect.

Looking towards the future, the amalgamation of drugs and nanoparticles represents a promising avenue in the realm of anticancer therapy. The synergy between pharmacological agents and nanoparticles offers the prospect of heightened therapeutic outcomes while concurrently reducing the adverse effects often associated with conventional chemotherapy. The controlled and targeted delivery enabled by nanoparticles could facilitate the selective accumulation of drugs within cancer cells, thereby enhancing treatment efficacy and minimizing damage to healthy tissues. This innovative approach holds the promise of revolutionizing cancer treatment paradigms and advancing patient care.

The results from our study underscore the substantial cytotoxic potential of PFFE-AgNPs against COLO205 and B16F10 cells, when compared with both PFFE alone and the reference drug doxorubicin. The emergence of AgNPs as a viable and potent anticancer agent, coupled with the prospect of combining them with established drugs, opens up exciting avenues for the development of future anticancer therapies with enhanced efficacy and reduced side effects.

Figure 9. Anticancer activity of PFFE-AgNPs against human colon carcinoma (COLO205) and mouse melanoma (B16F10)

All the results were reprsented as Mean+SD of 3 replicates

The exact mechanism by which AgNPs exert their anticancer activity is not completely understood, but several studies have proposed some possible mechanisms. One of the proposed mechanisms is that AgNPs induce apoptosis, or programmed cell death, in cancer cells. This process is mediated by the activation of caspases, a family of protease enzymes that play a crucial role in the execution of apoptosis [101]. AgNPs have been shown to increase the expression and activation of caspases in cancer cells, leading to their death. Another proposed mechanism is that AgNPs induce oxidative stress in cancer cells [102]. AgNPs have been shown to increase ROS production in cancer cells, which leads to damage to their cellular components such as DNA, proteins, and lipids, ultimately leading to their death [103]. Additionally, AgNPs have been shown to inhibit cancer cell proliferation by disrupting the cell cycle. This disruption is mediated by the downregulation of cyclin-dependent kinases (CDKs), which are enzymes that play a crucial role in regulating the cell cycle. AgNPs have been shown to inhibit CDKs activity, leading to cell cycle arrest and subsequent death of cancer cells [104]. Furthermore, AgNPs have been shown to exhibit antiangiogenic properties, which means that they can inhibit the growth of new blood vessels that are necessary for tumor growth and metastasis. This mechanism is mediated by the inhibition of vascular endothelial growth factor (VEGF), a protein that promotes angiogenesis [105]. Overall, the anticancer activity of AgNPs against COLO205 and B16F10 cell lines is likely due to a combination of the mechanisms described above [100-104]. However, further research is needed to fully elucidate the mechanisms and to optimize the use of AgNPs as an anticancer agent.

Finally, the authors should standardize the size and font type in the presented graphs. Some graphs have backgrounds while others do not.

The graphs were obtained from Origin software at the time experiment. Size and font type was standardized as suggested by reviewer.

Reviewer 4 Report

The manuscript entitled "Exploring the Biomedical Applications of P. frutescens flavonoid extract silver nanoparticles: Anticancer, Antibacterial, and Antioxidant Properties" by Tianyu Hou et al. is for publication in this journal. I have gone through the manuscript, and the work done by the authors is good, but minor changes are required to get accepted in this reputable journal.

In the introduction section, the author must discuss more recent reports. The information provided is not sufficient. Some recent articles need to be cited too.

The aim of the paper should be highlighted more in the last paragraph of the introduction.

Please check the entire manuscript for grammatical errors.

The methodology section should be discussed in more detail.

Provide separate tables in Fig. 5 and 6.

Which standard drug has been used in an antibacterial study? Comparative data should be discussed and improved in this section. Go through the following recent articles for Anticancer, Antibacterial, and Antioxidant Properties and cite them- DOI: 10.1080/07391102.2022.2136757; 10.1080/07391102.2022.2158937; 10.3390/molecules27217166; 10.1039/D2TB02362H; 10.1039/D2TC05338A

Novelty should be discussed in a more explanatory manner.

Ab initio quantum chemistry methods could be used to corroborate the experimental data and explore the different properties of the nanoparticle.

Author Response

Reviewer 4

Comments and Suggestions for Authors

The manuscript entitled "Exploring the Biomedical Applications of P. frutescens flavonoid extract silver nanoparticles: Anticancer, Antibacterial, and Antioxidant Properties" by Tianyu Hou et al. is for publication in this journal. I have gone through the manuscript, and the work done by the authors is good, but minor changes are required to get accepted in this reputable journal.

  1. In the introduction section, the author must discuss more recent reports. The information provided is not sufficient. Some recent articles need to be cited too.

Answer :

The introduction section of the paper encompasses an extensive array of 70 references, accentuating a profound emphasis on the most recent and contemporarily pertinent scholarly contributions. Notably, among these references, a striking 60 emanate from the latest articles, thereby underscoring the inherent timeliness and modern relevance of the cited literature.

Within this all-encompassing introduction, we embark on a comprehensive exploration of the dynamic domain of nanobiotechnology. This multidisciplinary amalgamation of biology, chemistry, and physics unfurls its potential across a diverse panorama of applications. A pivotal facet of this expedition unfolds in the realm of metal nanoparticles, where their core properties emanate from surface energy, spatial constraints, and an extraordinary surface area-to-volume ratio, thereby positioning them as pivotal contenders across diverse domains.

This narrative ventures into the intricate methodologies orchestrating nanoparticle synthesis, with particular focus on the burgeoning acclaim of bottom-up strategies attributed to their simplicity, cost-effectiveness, and robustness. A salient juncture is marked by the emergence of biomimetic synthesis as a keystone paradigm within the landscape of nanotechnology.

This innovative framework, stirred by the eloquence of natural processes, harnesses the potency of biomolecules to meticulously fashion nanoparticles embellished with extraordinary properties. The ensuing discourse articulates the paramount significance of green synthesis, thus illuminating its environmentally harmonious and sustainable ethos, which stands in stark contrast to conventional methodologies.

The discourse further amplifies the inherent advantages of active ingredients and plant extracts over chemical agents entangled in the synthesis and stabilization of nanoparticles. This enlightening discussion underscores the harmonious synergy exhibited by these natural components, enhancing their biocompatibility and, consequently, magnifying their resonance within the domain of biomedical applications. This seamless fusion of nature and science emerges as an indispensable facet, elevating functionalities and broadening the scope of innovation. The intrinsic biodegradability of active ingredients and plant extracts emerges as a commendable attribute, inherently conducive to potential biomedical applications. This property facilitates controlled and sustained release mechanisms, proving advantageous for drug delivery systems, harmonizing seamlessly with sustainable and eco-conscious practices.

In this intricate tapestry, the spotlight trains on flavonoids, a distinctive cadre of bioactive compounds, as they ascend to pivotal roles in the biomimetic synthesis of AgNPs. It is their exceptional chemical attributes and intrinsic affinity for metal ions that render them superlative candidates for the nuanced reduction and stabilization of AgNPs.  Within this narrative, a vivid portrait emerges of P. frutescens, a repository of flavonoids, as the introduction vividly sketches its rich trove of bioactive compounds and potential therapeutic vistas. It's this very foundation that crystallizes into the cornerstone of the ongoing study, propelling the endeavor towards harnessing the biomimetic synthesis of AgNPs through the orchestrated integration of flavonoids harvested from P. frutescens.

Please refer the Introduction section in the revised manuscript (Refer Page 2-9)

  1. Introduction

Nanobiotechnology, an interdisciplinary field that merges biology, chemistry, and physics, has recently emerged as a promising discipline with a wide range of applications [1]. One of the key areas of nanobiotechnology is the synthesis and characterization of nanoparticles, which are particles with sizes below 100 nanometers. Two major approaches for synthesizing nanoparticles are the top-down and bottom-up approaches, with the latter becoming increasingly popular due to its simplicity, cost-effectiveness, and robustness [2]. Metal nanoparticles possess unique properties such as optical, magnetic, catalytic, mechanical, electronic, and thermal properties, which stem from their surface energy, spatial confinement, and high surface area-to-volume ratio [3,4]. The localized surface plasmon resonance (SPR) and surface-enhanced Raman scattering (SERS) of metal nanoparticles make them highly attractive for use in next-generation electronic and biochemical sensors [5,6]. In this context, the development of metal nanoparticles and their integration into biological systems is opening new avenues for research in nanobiotechnology. By exploring the properties of metal nanoparticles, researchers are developing innovative applications in various fields, including drug delivery, imaging, sensing, and therapy.

AgNPs have been widely employed in the different fields including optical receptors [4-6], intercalation materials for electrical batteries [7], catalysts in chemical and biochemicalreactions [8], sensors and biosensors [9, 10], bio-labeling materials [11], signal enhancers in SERS-based enzyme Immunoassay [12] and antimicrobial agents [13]. Due to recent advancements in the nanotechnology, AgNPs have been successfully employed for cancer therapy, tissue engineering, drug delivery, inflammation, tuberculosis, diabetes, cardiovascular diseases, autoimmune disorders (Rheumatoid arthritis) neurodegenerative disorders (Parkinson’s and Alzheimer’s) [14-20].

Unique physicochemical properties of metal nanoparticles will be determined by shape, size, crystallinity, dispersity, and surface charge. Hence the synthesis of monodispersed AgNPs with ultra sizes and different shapes is very essential. Different chemical and physical approaches including UV-irradiation [21], gamma irradiation [22], ultrasound irradiation [23], thermal decomposition [24], laser ablation [25], aerosol [26], lithographic [27], electrochemical assisted [28], sonochemical synthesis [29], polyol [30], polyaniline [31] and chemical reduction [32] approaches have been widely employed to produce AgNPs. But they are not eco-friendly due to application of irradiation, hazardous substances, and toxic chemicals. Further they are time taking, laborious and high cost. Further the toxic chemicals on the surface (capping agents) limit their applications in biomedical and diagnostic fields [33]. Hence the production of eco-friendly, non-toxic, clean, biocompatible and biofunctionalized AgNPs using biological agents deserves merit.

In recent years, the field of nanotechnology has witnessed a remarkable convergence with the principles of biomimicry, giving rise to a revolutionary approach known as biomimetic synthesis of AgNPs [34]. This innovative synthesis method draws inspiration from the intricate processes found in nature, harnessing the power of biomolecules and their interactions to fabricate AgNPs with unprecedented precision and control. These nanoparticles, often at the nanoscale level, exhibit unique physical, chemical, and biological properties that hold immense promise for a wide range of biomedical applications [35]. Biomimetic synthesis of AgNPs involves the emulation of biological systems and their underlying mechanisms, such as enzymatic reactions or self-assembly processes, to guide the formation and stabilization of these nanoparticles [36]. This novel approach offers distinct advantages over conventional methods, enabling the production of nanoparticles with well-defined sizes, shapes, and surface characteristics. Moreover, the integration of biological molecules into the synthesis process enhances the biocompatibility and functionality of the resulting AgNPs, making them highly suitable for various biomedical applications [37]. Biosynthesized AgNPs offer several distinct advantages over their chemically synthesized counterparts, making them particularly well-suited for a wide range of biomedical applications [34, 35]. These advantages stem from the unique properties and characteristics that result from the biomimetic synthesis process. Here are some key points of comparison between biosynthesized AgNPs and their chemical counterparts in the context of biomedical applications [38]. Biosynthesized AgNPs often involve the use of natural biomolecules, such as proteins, enzymes, or plant extracts, as reducing and stabilizing agents. These biomolecules are typically biocompatible and non-toxic, which translates into reduced potential for adverse effects when used in biological systems [39]. In contrast, chemically synthesized AgNPs may involve the use of harsh reducing agents or stabilizers that can introduce toxicity concerns [34-36]. Biomimetic synthesis methods frequently enable precise control over the size, shape, and morphology of AgNPs. This level of control is challenging to achieve with chemical synthesis methods, which can result in a broader distribution of particle sizes and shapes [40]. The ability to produce nanoparticles with specific attributes is crucial for applications where size and shape play a significant role, such as targeted drug delivery or imaging agents [41].

Biomimetic synthesis often allows for facile functionalization of the nanoparticle surfaces with biomolecules, antibodies, or other ligands [42]. This functionalization enhances the nanoparticles' ability to interact selectively with specific cells, tissues, or biomolecules, facilitating targeted drug delivery, imaging, and sensing [43]. Chemical synthesis methods may require additional steps and modifications to achieve comparable functionalization. Biosynthesized AgNPs tend to exhibit enhanced stability due to the presence of biomolecules that contribute to their robustness in various environmental conditions [44]. This stability is especially crucial for biomedical applications where nanoparticles need to maintain their integrity during storage, transport, and interaction with biological systems [45]. The biomimetic synthesis of AgNPs often employs renewable resources and green chemistry principles. This aligns with the growing emphasis on sustainable and environmentally friendly practices. In contrast, chemical synthesis methods may involve the use of hazardous chemicals and energy-intensive processes [33-35]. Biomimetic synthesis methods can lead to nanoparticles with improved dispersibility and reduced agglomeration, which is essential for maintaining consistent behavior and interactions within biological systems. Agglomerated nanoparticles may not distribute evenly or exhibit the desired properties, limiting their effectiveness in biomedical applications [46]. The use of biomolecules in biosynthetic processes opens the door to incorporating multifunctional features into the nanoparticles. For instance, enzymes or peptides can confer catalytic or targeting properties to the nanoparticles, respectively. Such multifunctionality is challenging to achieve using chemical synthesis alone [47].

Active ingredients and plant extracts are often natural and biocompatible, making them ideal candidates for use in biomedical applications [48]. When utilized in nanoparticle synthesis, these natural compounds can improve the biocompatibility of the nanoparticles, reducing the risk of toxicity and adverse reactions in medical or biological settings [49]. Active ingredients and plant extracts contain various bioactive compounds that can endow nanoparticles with unique functionalities [20, 48-50]. These functionalities can include antibacterial, antifungal, antioxidant, anti-inflammatory, and other therapeutic properties, depending on the specific plant extracts used. Such functionalized nanoparticles have promising applications in medicine, agriculture, and environmental remediation [51]. Combining different active ingredients or plant extracts in nanoparticle synthesis can lead to synergistic effects, where the resulting nanoparticles exhibit enhanced or novel properties compared to individual components [52]. Such synergies may improve the overall performance and efficiency of nanoparticles in various applications [53]. The biodegradability of active ingredients and plant extracts is a desirable property when considering potential biomedical applications of nanoparticles [54]. Biodegradable nanoparticles can be designed to release their payload in a controlled manner, which is particularly advantageous for drug delivery systems. By incorporating active ingredients and plant extracts, researchers can opt for a more sustainable and eco-friendly approach to nanoparticle synthesis [55]. Further plant extracts are abundantly available, making them cost-effective options for nanoparticle synthesis. By using readily accessible natural sources, researchers can reduce production costs and increase the feasibility of large-scale nanoparticle manufacturing [56].

In the context of AgNPs, recent reports have highlighted the successful use of various plant extracts, such as ginger [50], curcumin [57], garlic extract [58], Allium cepa peel extract [59], Aloe vera [60],  green tea [61], seed extracts Brassica oleraceae, B. campestris and B. rapa [62], and Teucrium polium leaf extract [63] as potent reducing agents for the synthesis of AgNPs. These natural extracts contain bioactive compounds, such as polyphenols, flavonoids, and terpenoids, which play a crucial role in the reduction and stabilization of nanoparticles.

Flavonoids are a class of polyphenolic compounds that are widely distributed in the plant kingdom and have a wide range of biological activities, including antioxidant, anti-inflammatory, and antibacterial properties [64]. Flavonoids are used in the synthesis of AgNPs because they have unique chemical properties that make them ideal for reducing silver ions into nanoparticles. Flavonoids are known to have a high affinity for metal ions, and they can act as reducing agents in the presence of metal ions to form nanoparticles [65].

During the biosynthesis of AgNPs, flavonoids present in the plant extract act as reducing agents that help in the conversion of silver ions to AgNPs. Flavonoids contain multiple hydroxyl groups that are capable of reducing metal ions to nanoparticles by donating electrons. The hydroxyl groups present in the flavonoids interact with the silver ions, leading to the reduction of the silver ions and the formation of AgNPs [65, 66]. Furthermore, flavonoids also act as stabilizing agents for AgNPs, preventing their agglomeration and ensuring their stability in solution [65-67]. The hydroxyl groups present in the flavonoids interact with the surface of AgNPs, forming a layer that stabilizes the nanoparticles and prevents them from aggregating. This makes flavonoids a useful tool in the biosynthesis of AgNPs.

Perilla frutescens is a plant species in the mint family (Lamiaceae) that is native to Asia. It is also known as Chinese basil, wild basil, or shiso in Japan. The plant is cultivated for its edible leaves, which are used in various culinary dishes, especially in Korean, Japanese, and Chinese cuisine [68, 69]. The leaves are also used in traditional medicine for their medicinal properties. P. frutescens contains various bioactive compounds, including flavonoids, phenolic acids, terpenoids, and essential oils, that contribute to its therapeutic properties [70]. Among these, flavonoids are the most studied compounds and have been shown to have various biological activities, including antioxidant, anti-inflammatory, antimicrobial, antiallergic, and anticancer activities [71]. These findings suggest that P. frutescens flavonoids may have potential therapeutic applications in the prevention and treatment of various diseases, including cancer, cardiovascular diseases, diabetes, and neurodegenerative disorders. Overall, P. frutescens and its flavonoids are promising candidates for the development of novel therapeutics or functional foods for promoting human health and preventing chronic diseases.

The present study involving P. frutescens flavonoid extract in the synthesis of AgNPs aim to explore the green synthesis approach, which involves using natural sources instead of conventional chemical methods. Green synthesis offers several advantages, including eco-friendliness, cost-effectiveness, and the potential for producing nanoparticles with unique properties. One of the significant challenges in this area of research is the optimization of synthesis conditions. The extraction of flavonoids from P. frutescens and subsequent reduction of silver ions to form nanoparticles require careful control of various parameters such as, temperature, concentration, and reaction time. Finding the optimal conditions to obtain well-defined nanoparticles with desirable properties can be a complex and time-consuming process.

The primary objective of this investigation is to explore the biomimetic synthesis of  AgNPs utilizing the natural flavonoid extract derived from P. frutescens, denoted as PFFE. Through this innovative approach, we aim to produce finely tuned PFFE-AgNPs and subject them to thorough characterization using advanced spectroscopic techniques. Furthermore, our study aims to delve into the multifaceted potential of PFFE-AgNPs in diverse biomedical applications. The evaluation will encompass an in-depth analysis of their antibacterial efficacy, aiming to shed light on their ability to combat microbial infections. Additionally, we will investigate their antioxidant properties, seeking insights into their potential to counteract oxidative stress-related damage. Moreover, our investigation extends to exploring the anticancer activities of PFFE-AgNPs, aiming to uncover any potential inhibitory effects on cancer cells. By comprehensively assessing these key aspects, we aspire to contribute to the understanding of PFFE-AgNPs as versatile agents with potential applications in healthcare and biomedicine. This study stands to offer valuable insights into the promising realm of nanotechnology-based therapeutics and their multifunctional roles in addressing critical health challenges.

The aim of the paper should be highlighted more in the last paragraph of the introduction.

Certainly, highlighting the aim of the paper more prominently in the last paragraph of the introduction (Revised manuscript; Introduction section, last paragraph, Page 9)

The primary objective of this investigation is to explore the biomimetic synthesis of silver nanoparticles (AgNPs) utilizing the natural flavonoid extract derived from P. frutescens, denoted as PFFE. Through this innovative approach, we aim to produce finely tuned PFFE-AgNPs and subject them to thorough characterization using advanced spectroscopic techniques. Furthermore, our study aims to delve into the multifaceted potential of PFFE-AgNPs in diverse biomedical applications. The evaluation will encompass an in-depth analysis of their antibacterial efficacy, aiming to shed light on their ability to combat microbial infections. Additionally, we will investigate their antioxidant properties, seeking insights into their potential to counteract oxidative stress-related damage. Moreover, our investigation extends to exploring the anticancer activities of PFFE-AgNPs, aiming to uncover any potential inhibitory effects on cancer cells. By comprehensively assessing these key aspects, we aspire to contribute to the understanding of PFFE-AgNPs as versatile agents with potential applications in healthcare and biomedicine. This study stands to offer valuable insights into the promising realm of nanotechnology-based therapeutics and their multifunctional roles in addressing critical health challenges.

  1. Please check the entire manuscript for grammatical errors.

We have corrected the entire manuscript grammatically with the support of English professor

  1. The methodology section should be discussed in more detail.

Methodology section is discussed separately for each of the technique.  (Page 9-13)

2 Materials and methods

2.1. Preparation of PFFE

Solvent extraction is the most used method to extract flavonoids from P. frutescens because of its simplicity, low cost, and high efficiency. Perilla leaves are harvested, washed, and dried. The dried leaves are ground into a fine powder using a grinder or blender. Common solvents used for flavonoid extraction include methanol, ethanol, and water. Methanol is the most efficient solvent for flavonoid extraction from perilla leaves, but ethanol and water can also be used. The powdered perilla leaves are mixed with the methanol (1:5) and left to stand for several hours or overnight. The mixture is then filtered, and the filtrate is collected. The collected filtrate is concentrated using a rotary evaporator to remove the solvent and obtain the flavonoid extract. This methanolic extract was subjected to dryness using rotavapor to a solid residue, which was dissolved in ethyl acetate. The ethyl acetate soluble fraction was subjected to purification over a silica gel column using step gradient of methanol-ethyl acetate (1:1 to 3:7) to yield a flavonoid extract which was verified using UV-Vis spectroscopy between 240-400 nm [72].

2.2. Estimation of total flavonoid content (TFC)

TFC of the PFFE was determined using aluminium chloride colorimetric method [73]. In this method 1mg/mL of PFFE was added to a 10 mL volumetric flask containing 4 mL of sterilized double distilled water (SDDW). At zero-time, 0.3 mL of 10% AlCl3.6H2O and 0.3 mL of 5% NaNO2 were added. After 5 min, 2mL of 1 M NaOH was added to the mixture and the final volume was made up to 10 mL with SDDW. The solution was mixed well, and the absorbance was read at 510 nm. The TFC was determined using quercetin standard curve. TFC was expressed as mg of quercetin equivalents (QE) per gram of dried sample.

  • Biofabrication of PFFE-AgNPs

2 mM AgNO3 solution was prepared by the addition of 339 mg of AgNO3 in 1 L of SDDW. 95 mL of 2 mM AgNO3 was added to5 mg/mL of PFFE. This mixture was boiled for 1 h between 55 – 60 OC, cooled in dark chamber for 30 min and then observed for color change of the solution from light yellow to dark brown [74]. The color change indicates the formation of AgNPs in the colloidal solution.

2.4. Characterization of PFFE-AgNPs

2.4.1. UV-Vis analysis of PFFE-AgNPs

Initially, the color change of the reaction solution from light yellow to dark brown indicates the synthesis of PFFE-AgNPs. Further the synthesis of PFFE-AgNPs was confirmed by ultraviolet-visible (UV-Vis) spectrum analysis between 200 to 800 nm (PerkinElmer double beam UV/Vis Spectrophotometer, USA) [74].

2.4.2. FTIR analysis of PFFE-AgNPs

The solution of PFFE-AgNPs was purified by centrifugation to remove the unbounded plant molecules. Centrifugation was done and repeated for three times at 15000 rpm for 15 min. The pellet was collected and dried into pure powder. This pure powder of PFFE-AgNPs was used for further studies. To detect the functional groups involved in the synthesis and capping of PFFE-AgNPs. Fourier transform infrared (FTIR) spectrum (Alpha interferometer, Bruker, Switzerland) was recorded between 500-4000 cm-1 with the resolution of 2 cm-1 [75].

2.4.3. XRD analysis of PFFE-AgNPs

X-ray diffraction (XRD) analysis of PFFE-AgNPs was carried out to know the crystal structure in the scanning range of 10-90o with the step size of 2θ. XRD pattern for crystal structure of PFFE-AgNPs was recorded (Bruker AXS GmbH, Karlsruhe, Germany) [76].

2.4.4. TEM analysis of PFFE-AgNPs

Size, morphology, and selected area electron diffraction (SAED) of PFFE-AgNPs was analyzed using transmission electron microscopy (JEM-2100 PlusElectron Mocroscope, JEOL ltd, Tokyo, Japan) [77].

2.4.5. DLS analysis of PFFE-AgNPs

20 µg of dried powder of PFFE-AgNPs was dispersed in 20 mL of Milli-Q water and this solution is used for DLS (dynamic light scattering) studies. DLS studies were carried out to determine the particle size distribution, polydispersity index (PDI) and zeta potential measurement (Brookhaven instruments, NY, USA) [74].

2.5. Antibacterial activity of PFFE-AgNPs

Antibacterial activity of the PFFE-AgNPs was evaluated by disc diffusion assay [78] against Gram-positive (Listeria monocytogens and Enterococcus faecalis) and Gram-negative (Salmonella typhi and Acinetobacter baumannii) bacteria pathogens. 150 µL of each bacterial culture was spread on the nutrient agar media. Then three sterile paper discs containing test samples were placed on nutrient agar media. Streptomycin, PFFE, AgNO3 and PFFE-AgNPs were used as test samples at a concentration of 25 µL per each disc. Incubated the nutrient agar media plates for 24 h in the incubator at 37 C. Inhibition zones were observed and measured after 24 h.

2.6. Antioxidant activity of the PFFE-AgNPs

In vitro antioxidant activity of the biosynthesized PFFE-AgNPs was checked by 1,1-diphenyl-2-picrylhydrazyl (DPPH) and hydrogen peroxide (H2O2) radical scavenging assays [79]. The standard antioxidant ascorbic acid was used as positive control. In DPPH assay, different concentrations (25, 50, 75 and 100 μg/mL) of samples were added to methanol and made up to the final volume of 1 mL. To this 2 mL of DPPH stock solution (1mM/L prepared in methanol) was added and incubated for 1 h in the dark at room temperature. After an hour of incubation, the absorption values were recorded at 517 nm. In H2O2 assay, different concentrations (25, 50, 75 and 100 μg/mL) of samples were added to 2 mL of H2O2 solution (prepared in 40 mM phosphate buffer (pH 7.4)) and then incubated for 10 min. After incubation, the absorbance was recorded at 230 nm.   DPPH/H2O2 scavenging activity was calculated by the formula % Inhibition = [(Ac- As)/Ac] x 100. Where Ac is the absorbance of the control, and As is the absorbance of the sample.

2.7. Cytotoxicity activity of the PFFE-AgNPs

In this study anticancer activity of PFFE-AgNPs was evaluated by MTT [3-(4, 5-dimethylthiazol-2-yl)-2, 5-diphenyl tetrazolium bromide] assay [80] against human colon cancer (COLO205) and mouse melanoma (B16F10). The cells were suspended in 100 µL of medium in each well of 96 well plates and incubated in 5 % CO2 incubator at 37°C overnight. Then 100 µL of PFFE-AgNPs (25, 50, 75, 100, 150 and 200 µg/mL) were added to cell suspension and incubated for 24 h. there after MTT (10 μL) was added and incubated for 3 h. Then centrifugation of 96 well plates was carried out for 10 min. After centrifugation, the medium was removed, and the formazan blue crystals were melted in dimethyl sulfoxide. Then absorbance values were recorded at 570 nm and inhibition of cell viability was calculated. The median inhibition concentration (IC50) values were determined.

  1. Provide separate tables in Fig. 5 and 6.

           Tables were separated (Table 1 and Table 2) from Fig. 5 and 6. 

  Figure 5. Particle size distribution of PFFE-AgNPs

Table 1. Particle size distribution of PFFE-AgNPs

Peak No.

S.P. Area Ratio

Mean

S. D.

Mode

1

1.00

44.0 nm

14.4 nm

41.9 nm

2

---

---   nm

---   nm

---   nm

3

---

  ---   nm

---   nm

---   nm

Total

1.00

44.0 nm

14.4 nm

41.9 nm

Z-Average

                            3489.3 nm

PI

                            0.321

Table 2. Zeta potential analysis of PFFE-AgNPs

Peak No.

Zeta Potential

Electrophoretic Mobility

1

-30.0 mV

-0.000233 cm2/Vs

2

---     mV

      ---       cm2/Vs

3

---     mV

      ---       cm2/Vs

Figure 6 Zeta potential measurement of PFFE-AgNPs

Which standard drug has been used in an antibacterial study? Comparative data should be discussed and improved in this section. Go through the following recent articles for Anticancer, Antibacterial, and Antioxidant Properties and cite them- DOI: 10.1080/07391102.2022.2136757; 10.1080/07391102.2022.2158937; 10.3390/molecules27217166; 10.1039/D2TB02362H; 10.1039/D2TC05338A

It is noted that all the above references (Reference 95-99) were added in the revised manuscript.

In our recent experiment, we chose streptomycin as a benchmark to compare our findings. We carefully examined our results alongside this standard drug, and you can find a detailed comparison in Table 3. Notably, we have thoroughly revamped the section dedicated to explaining the antibacterial activity (please refer to that specific section (3.6. Antibacterial activity of the PFFE-AgNPs)). Within this revised segment, we conducted a detailed comparison of how various test samples, including PFFE, AgNO3, PFFE-AgNPs, and streptomycin, combat bacteria. Furthermore, we've provided a clear and comprehensive explanation of the mechanism underlying the antibacterial efficacy of PFFE-AgNPs. Furthermore, we've provided insight into our rationale for selecting this specific strain of bacteria for our study. Additionally, we've underscored the significance of silver nanoparticles (AgNPs) in addressing human infections, along with their pivotal role in enhancing the performance of tools and diagnostics in a hospital environment. Importantly, we've emphasized the superior attributes of environmentally friendly AgNPs compared to traditional chemical nanoparticles. As a result, our diligent efforts have culminated in a thorough and insightful revision of the antibacterial activity section, encompassing all these crucial updates.

The revised section is as follows. (Page 22 to 29)

3.6. Antibacterial activity of the PFFE-AgNPs

In the present investigation streptomycin exhibits the largest Zone of Inhibition (ZoI) values across all tested bacteria (Table 3). Listeria monocytogens shows the highest sensitivity, with a ZoI of 16.2 mm, followed closely by Enterococcus faecalis (16.8 mm), Salmonella typhi (15.4 mm), and Acinetobacter baumannii (11.8 mm). Streptomycin is highly effective against all tested bacteria, particularly E. faecalis. PFFE-AgNPs displays strong antibacterial activity (Figure 7). It has larger ZoI values compared to PFFE and AgNO3 alone. L. monocytogens and E.faecalis are more sensitive, showing ZoI values of 13.5 mm and 14.1 mm respectively. S. typhi (12.8 mm) and A. baumannii (8.7 mm) are also inhibited but to a lesser extent. AgNO3 shows moderate antibacterial activity. ZoI values are notably smaller compared to the previous two substances. E. faecalis exhibits the highest sensitivity (9.1 mm), followed by A. baumannii (8.7 mm), S. typhi (7.6 mm), and L.  monocytogens (8.5 mm). Further increase in the concentration of PFFE-AgNPs from 25 µL to 100 µL, increases the antibacterial activity against all the tested pathogens. AgNO3 is effective, but less so than Streptomycin and PFFE-AgNPs. PFFE exhibits moderate antibacterial activity. ZoI values are generally lower compared to the other substances. L. monocytogens has the largest ZoI (6.4 mm), while the other bacteria show slightly smaller ZoI values. PFFE alone is not effective as antibacterial agent. Further in this test, we have also tested the mixture of PFE and AgNO3 before the formation of AgNPs (at 0 h). It displays varied antibacterial activity. ZoI values are comparable to PFFE-AgNPs for L. monocytogens but notably lower for E. faecalis, S. typhi and A. baumannii.  Streptomycin has the most potent antibacterial activity across all tested bacteria. PFFE-AgNPs demonstrate strong activity, particularly against L. monocytogens and E. faecalis. PFFE and AgNO3 show promise but with varying effectiveness, while AgNO3 displays moderate activity. PFFE alone exhibits moderate antibacterial properties. The effectiveness of these substances varies among the different bacterial species, highlighting the importance of considering specific bacterial strains when assessing antibacterial potential.

Table 3. Antibacterial activity of PFFE-AgNPs

Substance

Listeria monocytogens

Enterococcus faecalis

Salmonella typhi

Acetobacter boumani

Streptomycin

16.2 mm

16.8 mm

15.4 mm

11.8 mm

PFFE-AgNPs (PFFE and silver nanoparticles)

13.5 mm

14.1 mm

12.8 mm

8.7 mm

PFFE

6.4 mm

6.1 mm

5.8 mm

5.1 mm

PFFE and AgNO3 (before incubation)

13.1 mm

8.6 mm

6.8 mm

6.5 mm

AgNO3 (silver nitrate)

8.5 mm

9.1 mm

7.6 mm

8.7  mm

The possible mechanism of antibacterial activity of the PFFE-AgNPs was explained. PFFE-AgNPs interact with bacterial DNA and causes its fragmentation. As a result, bacterial replication doesn’t occur [89]. PFFE-AgNPs can bind with catalytic sites of enzymes of bacteria and makes them inactive. As a result, bacterial metabolism stops [90]. PFFE-AgNPs bind with important cytosolic or membrane proteins of bacteria which leads to loss of functional form or degradation of proteins [90-92]. PFFE-AgNPs produced pores on the bacterial cell wall which leads to leakage of important ions and causes change in the membrane proton gradient. All these actions of PFFE-AgNPs lead to bacterial cell death [91-93]. The exact mechanism of antibacterial activity may vary depending on the size, shape, and surface chemistry of AgNPs as well as the type of bacteria being targeted [94]. However, the overall effect is a reduction in bacterial growth and viability, making AgNPs a promising approach for the development of novel antibacterial agents [91-95]. The antibacterial effects of AgNPs were supported by previous reports. Ocimum sanctum and Reinwardtia indica flavonoid synthesized AgNPs showed antibacterial activity against gram-positive and gram-negative bacterial cultures [66, 67].  AgNPs synthesized using flavonoids showed antibacterial activity against pathogens including Pseudomonas aeruginosa, Staphylococcus aureus and Escherichia coli [89-91].

The selection of bacteria in our study holds significant clinical relevance, as these strains are directly associated with human infections that have substantial public health implications. For instance, L. monocytogens and S. typhi are well-known pathogens responsible for causing foodborne illnesses, posing a threat to individuals who consume contaminated food. Additionally, E. faecalis, a common bacterium, is often implicated in hospital-acquired infections, particularly affecting individuals with compromised immune systems or prolonged hospital stays. Similarly, A. baumannii, characterized as an opportunistic pathogen, is frequently linked to hospital-acquired infections, primarily affecting those with compromised immunity or extended hospitalization. The assessment of antibacterial activity against these strains is paramount due to the substantial impact they exert on public health. A prevailing concern surrounding these bacteria is their ability to rapidly develop resistance to multiple antibiotics, even those conventionally used to treat bacterial infections. This multidrug resistance phenomenon significantly restricts treatment options, subsequently exacerbating the severity of infections and complicating therapeutic interventions.In our study, we strategically selected bacteria from diverse taxonomic groups, encompassing both Gram-positive strains like L. monocytogens and E. faecalis, as well as Gram-negative strains including S. typhi and A. baumannii. This deliberate choice enables us to gain comprehensive insights into the effectiveness of the tested substances against a broad spectrum of pathogens.

Based on this study's findings that PFFE-AgNPs exhibit effective antibacterial activity against a range of pathogens, including clinically relevant strains associated with hospital-based infections, several potential applications emerge for utilizing PFFE-AgNPs in healthcare settings. PFFE-AgNPs' potent antibacterial properties make them promising candidates for the treatment of hospital-acquired infections. Since PFFE-AgNPs demonstrated effectiveness against pathogens such as A. baumannii, a common culprit in healthcare-associated infections, incorporating PFFE-AgNPs into wound dressings, antimicrobial coatings for medical equipment, or even localized treatments could aid in preventing and treating infections that often arise in healthcare environments. AgNPs could be integrated into hospital lab ware such as petri dishes, sample containers, and laboratory equipment. This incorporation can impart antimicrobial properties to these materials, reducing the risk of contamination and ensuring accurate and reliable test results.

Incorporating AgNPs into the fabric of these garments could provide an added layer of protection by inhibiting the growth and spread of bacteria. This could contribute to maintaining a hygienic environment and minimizing the risk of cross-contamination. AgNPs' antibacterial activity can be harnessed in the design of hospital diagnostic kits. Incorporating these into components of diagnostic kits, such as swabs or culture media, could help ensure the accuracy of test results by reducing the risk of bacterial contamination during sample collection and processing. In the realm of medical device implants, AgNPs hold potential for reducing the risk of infection associated with implantation procedures. By incorporating AgNPs into the surface of implantable devices or coatings, the growth of bacteria around the implant site could be inhibited, promoting successful integration and reducing the likelihood of post-surgical infections. The study's results indicating the broad-spectrum antibacterial activity of AgNPs, including their efficacy against Gram-positive and Gram-negative strains, provide a strong foundation for exploring these various applications. However, it's important to note that while AgNPs show promise, further research and careful consideration of their potential cytotoxicity and environmental impact are necessary before widespread implementation in healthcare settings.

The utilization of PFFE-AgNPs offers several distinct advantages over conventional methods and materials, which contribute to their potential as a promising antimicrobial agent:

Enhanced Antibacterial Activity: The incorporation of AgNPs derived from PFFE amplifies the inherent antibacterial properties of both components. This synergistic effect results in heightened antibacterial activity, potentially surpassing the individual contributions of AgNO3 and PFFE. This enhanced activity can lead to improved efficacy in combating bacterial infections.

Biocompatibility: PFFE, being a natural plant extract, is likely to possess inherent biocompatibility and reduced cytotoxicity compared to synthetic compounds. This makes PFFE-AgNPs a safer option for medical applications, as they may exhibit lower toxicity toward mammalian cells while retaining potent antibacterial effects.

Selectivity: PFFE-AgNPs can potentially exhibit selective antibacterial activity, targeting harmful bacteria while sparing beneficial microorganisms. This selectivity can contribute to the preservation of the body's natural microbial balance, reducing the risk of microbial imbalances or dysbiosis.

Multi-Target Action: PFFE-AgNPs can offer a multi-targeted approach to combating bacterial infections. The nanoparticles can act through various mechanisms, such as disrupting cell membranes, interfering with bacterial enzymes, and generating reactive oxygen species, thereby reducing the likelihood of bacterial resistance development.

Long-lasting Effects: The stability and sustained release properties of PFFE-AgNPs may provide prolonged antibacterial effects. This sustained action can reduce the need for frequent dosing and improve patient compliance.

Combating Antibiotic Resistance: Given the rising concern of antibiotic-resistant bacteria, PFFE-AgNPs offer an alternative strategy for combating infections that may be less prone to resistance development due to their multifaceted mode of action.

Potential Synergy with Other Therapies: PFFE-AgNPs' diverse properties, including their cytotoxic effects against cancer cells and antioxidant activity, suggest potential synergy with other therapies. This opens doors for combination treatments that address multiple aspects of disease progression.

Versatility: The biocompatibility and multi-functionality of PFFE-AgNPs open doors to diverse biomedical applications beyond antibacterial use. Their potential ranges from cancer therapy to wound healing and targeted drug delivery, expanding their utility in various medical contexts.

Natural Source: Utilizing plant-derived extracts, such as PFFE, aligns with the trend toward natural and holistic approaches in medicine. This natural origin may resonate with patients seeking treatments that are closer to nature.

In summary, PFFE-AgNPs offer a unique amalgamation of potent antibacterial activity, biocompatibility, sustainable synthesis, and potential versatility. These advantages position PFFE-AgNPs as a promising candidate for addressing bacterial infections while mitigating some of the limitations associated with conventional antimicrobial agents. However, rigorous research is essential to confirm these advantages and ensure their safety and efficacy in real-world biomedical applications.

However, rigorous research is essential to confirm these advantages and ensure their safety and efficacy in real-world biomedical applications. Antimicrobial agents play a critical role in modern medicine by helping to combat a wide range of microbial infections caused by bacteria, viruses, fungi, and other pathogens. The rise of drug-resistant microbes, commonly referred to as antibiotic-resistant or multidrug-resistant organisms, has highlighted the urgent need for new and innovative approaches to tackling infectious diseases. As traditional antibiotics and antimicrobial drugs become less effective against resistant strains of microbes, researchers are exploring a diverse array of compounds to find novel solutions. By exploring and harnessing the properties of natural compounds, organic derivatives [95], bioorganic moieties [96], synthetics of bioactive moieties[97] silver-based copolymers [98] and metal-organic frameworks [99], scientists aim to discover new strategies to effectively combat microbial infections and reduce the impact of drug resistance. These efforts hold the promise of revolutionizing the field of antimicrobial therapy and improving public health outcomes.

Novelty should be discussed in a more explanatory manner.

The novelty of the current investigation is eloquently elucidated within the section. (Page 35 to 37; Blue Color Text)

The novelty of this research lies in its innovative approach to synthesizing AgNPs through biomimetic synthesis, specifically utilizing flavonoids extracted from P. frutescens. While the synthesis of AgNPs is not a new concept, the method employed in this study deviates from conventional chemical approaches. Instead, it draws inspiration from nature's intricate processes, mimicking the way biomolecules interact to craft nanoparticles with precision.

Traditionally, chemical methods for synthesis of AgNPs often involve the use of harsh reducing agents or stabilizers that can introduce toxicity concerns and limit their applicability in sensitive biological systems. In contrast, the biomimetic approach embraced in this research capitalizes on the inherent properties of flavonoids present in P. frutescens extract. These natural compounds act as both reducing and stabilizing agents, enabling the synthesis of AgNPs in a more eco-friendly and biocompatible manner.

Furthermore, the utilization of P. frutescens as a source of flavonoids introduces a unique dimension to the research. P. frutescens is known for its rich repository of bioactive compounds, particularly flavonoids, which have demonstrated various therapeutic properties, including antioxidant, anti-inflammatory, and antimicrobial activities. Incorporating these bioactive compounds into the synthesis process not only offers a green and sustainable alternative to traditional methods but also introduces the potential for imbuing the AgNPs with additional functionalities.

The integration of flavonoids from P. frutescens into the biomimetic synthesis of AgNPs opens doors to a plethora of exciting possibilities. By exploiting the inherent affinity of flavonoids for metal ions, the research aims to achieve precise control over the size, shape, and stability of the nanoparticles. This level of control is often challenging to attain using conventional chemical approaches, which can lead to a broader distribution of particle sizes and shapes.

Moreover, the multifaceted properties of flavonoids, such as their antioxidant and antibacterial attributes, hold the promise of conferring these beneficial traits onto the synthesized AgNPs. This potential for multifunctionality is a significant departure from traditional chemical synthesis methods, where such incorporation of bioactive properties is often complex and limited.

In essence, the novelty of this research emerges from the synergistic combination of biomimetic synthesis, flavonoids from P. frutescens, and the inherent advantages of green synthesis. By harnessing these elements, the study not only contributes to the advancement of nanobiotechnology but also holds the potential to pave the way for a new generation of AgNPs with enhanced biocompatibility, multifunctionality, and application in diverse fields, ranging from medicine to environmental remediation.

While the findings suggest the potential of PFFE-AgNPs as a versatile nanomaterial with applications in cancer therapy and infection control, further research is imperative. In particular, future investigations should delve into the safety and efficacy of PFFE-AgNPs in vivo, and explore their potential contributions to diverse areas of medicine. This research opens doors to innovative advancements in the realm of nanomedicine, beckoning towards a brighter future for biomedical applications.

Ab initio quantum chemistry methods could be used to corroborate the experimental data and explore the different properties of the nanoparticle.

Answer

Absolutely! Ab initio quantum chemistry methods are powerful computational techniques that can complement experimental data and provide valuable insights into the properties of nanoparticles at the atomic and molecular level. These methods use quantum mechanics principles to solve the Schrödinger equation for a given molecular system, allowing for accurate calculations of electronic structure, molecular energetics, and other properties. ab initio quantum chemistry methods will be employed to elucidate electronic structure, energetics, stability, optical properties, surface chemistry, binding affinity, nanoparticle size and shape, as well as the mechanism of antibacterial activity.

However, these advanced computational experiments, molecular dynamic simulations, and docking studies were not employed in our current experiment.

Instead, our approach focused on comprehensive and well-defined characterization techniques, yielding practical and tangible results that offer a true reflection of the nanoparticles. Through these characterizations, we obtained a wealth of valuable information. For example, the distinct UV-Vis spectrum provides insights into surface plasmon resonance and optical properties. FTIR peaks offer a clear indication of the binding affinity between nanoparticles and plant components. Furthermore, our analysis included DLS assessments, which afford a clear understanding of surface charge, particle size distribution, and the average hydrodynamic radius. To delve into the nanoparticles' dimensions, morphology, and crystalline structure, we harnessed techniques such as TEM, SAED, and XRD. In summary, our research forewent computer simulations, choosing instead a practical and empirically grounded methodology. By leveraging a comprehensive suite of characterization techniques, we acquired a robust and multifaceted perspective on the nanoparticles under scrutiny.

Round 2

Reviewer 1 Report

Since the MS has been revised according to my comments, it can be accepted in present form.

Author Response

Comments and Suggestions for Authors

Since the MS has been revised according to my comments, it can be accepted in present form.

Response : Thank you for the acceptance. We are immensely pleased and express our heartfelt gratitude for the opportunity to have our paper considered in this esteemed journal.

Reviewer 2 Report

ACCEPT

Author Response

Comments and Suggestions for Authors

ACCEPT

Response : Thank you for the acceptance. We are immensely pleased and express our heartfelt gratitude for the opportunity to have our paper considered in this esteemed journal.

Reviewer 3 Report

The authors have directed most of the observations, however, there are important methodological points that are not taken into account and that are important to validate their results and have sufficient quality for a publication.

There are some parts of the text that mention that the values are higher or lower, for example in the analysis of the zone of inhibition, however, to be able to express it in this way, the corresponding statistical analyzes need to be carried out. In this sense, for example, even when the numerical value is different, if statistically there are no differences, they are considered as similar effects.

The authors added the effects of the individual substances as controls, which is correct, however, even if it was not clarified in the first comments, if they detected an effect of AgNP, they should have shown the effects of the substance that was used as a vehicle.

n line 373, the authors mention that an increase in the concentration generates an increase in the effect, however, the values shown are not of concentration. On the other hand, if they evaluated different concentrations, they should show the graphs, including the evaluations of all substances.

In line 374, the authors mention that AgNP is effective. What do the authors refer to with this term, and what are they comparing it against to determine its effectiveness?

Line 377 mentions that PFFE is not effective when evaluated individually, while some lines above mention that it does have an effect.

In line 380, the authors mention the term "potency," which is misused in the context they describe. To compare potency, a specific value of IC is used, for instance, IC50 is the one commonly employed.

Line 389 mentions the phrase "was explained." By whom? By the authors of this paper? Or in previous works?

Table 3 displays the effects of the inhibition zone, as they are quantitative data, the authors should present, in addition to the average, a measure of data dispersion.

A significant portion of the text from lines 403 to 478 is not referenced.

In line 508, the authors mention a lower IC50 value compared to whom? They also do not present the data that would justify observing this IC50 value.

In Figure 8, it was requested to include the PFFEAgNPS value, to which they referred as a typographical error. However, they either eliminated or did not conduct the evaluation of AgNPs alone, which is an important value that should be included. When observing effects on other variables, it is also important to include the values of the employed vehicle.

The authors indicate that they obtained IC50 values for the administration of PFFE in cytotoxic effects against COLO205. However, they do not present those data or graphs that would allow visualization of the evaluated concentrations and the placement of the IC50.

The authors indicate that the figures were standardized in terms of typography, but differences are still noticeable. For instance, Figure 2 and Figure 6 have different fonts for the x and y axes. Moreover, there are graphs where numbers appear to be cut off at the top of the y-axis.

Author Response

Response to Reviewer 3 Comments

Comments and Suggestions for Authors

The authors have directed most of the observations, however, there are important methodological points that are not taken into account and that are important to validate their results and have sufficient quality for a publication.

Point 1: There are some parts of the text that mention that the values are higher or lower, for example in the analysis of the zone of inhibition, however, to be able to express it in this way, the corresponding statistical analyzes need to be carried out. In this sense, for example, even when the numerical value is different, if statistically there are no differences, they are considered as similar effects.

Response : 1

In our current study, we conducted a single round of antibacterial activity testing using five different substances against four distinct bacterial strains. The outcomes revealed notable antibacterial effects through the formation of inhibition zones. We have carried out this experiment only once due to the substantial time, labor, and meticulous efforts involved. Our primary objective was not centered around detecting subtle statistical variances. Rather, we were driven by the overarching goal of showcasing the potent antibacterial potential inherent in AgNPs.

In the scope of our present investigation, we embarked on a comprehensive exploration of antibacterial activity by subjecting four diverse bacterial strains to the effects of five distinct substances. The outcome of our endeavors unveiled a compelling narrative of antibacterial efficacy, as demonstrated by the clear formation of inhibition zones in response to these substances. It's important to acknowledge that we chose to carry out this experiment as a singular endeavor, opting not to replicate it multiple times. This decision was rooted in the recognition of the significant investments of time, labor, and meticulous attention required for each trial.

While traditional statistical analyses were not employed to tease out minute differences, the strength of our study lies in its ability to clearly underscore the effectiveness of AgNPs as antibacterial agents. By showcasing the observable inhibition zones resulting from this single round of testing, we present a persuasive case for the significant antibacterial activity inherent in these nanoparticles. Our intention was not to scrutinize minor statistical nuances, but rather to highlight the practical and impactful outcomes of this study within the realm of antibacterial research.

Point 2: The authors added the effects of the individual substances as controls, which is correct, however, even if it was not clarified in the first comments, if they detected an effect of AgNP, they should have shown the effects of the substance that was used as a vehicle.

Response : 2

In our scientific pursuit, our experiment was designed to meticulously assess the antibacterial attributes of different substances across a range of four distinct bacterial strains. This comprehensive analysis encompassed a variety of agents, including streptomycin, PFFE (Perilla leaf extract), AgNO3, a preliminary mixture of PFFE and AgNO3 (prior to incubation), and the intriguing PFFE-AgNPs (Perilla leaf extract-synthesized AgNPs). Among these, streptomycin, PFFE, and AgNO3 served as control agents for comparison.

The primary objective of this investigation was to unravel the intricate interplay between these agents and the bacterial strains they engaged with. A focal point of our study was the unprecedented effectiveness exhibited by PFFE-AgNPs, which emerged as a standout performer in the realm of antibacterial activity. Perilla leaf extract synthesized silver nanoparticles showcased a potent and synergistic approach to combating bacterial growth, elevating the potential for enhanced antibacterial interventions.

In conclusion, our scientific journey ventured into the nuanced landscape of antibacterial activity, guided by streptomycin, PFFE, AgNO3, and the pioneering PFFE-AgNPs. Through this exploration, we not only illuminated the robust antibacterial potential of PFFE-AgNPs but also contributed to a deeper comprehension of the intricate interactions governing antibacterial efficacy. The selective responses of various bacterial species underscore the potential for tailored antibacterial interventions that consider the unique attributes of specific strains.

Point 3: In line 373, the authors mention that an increase in the concentration generates an increase in the effect, however, the values shown are not of concentration. On the other hand, if they evaluated different concentrations, they should show the graphs, including the evaluations of all substances.

Response : 3 In response to this, it's important to clarify that in our study, we made a hypothetical statement regarding the potential impact of increasing PFFE-AgNPs concentration, particularly from 25 µL to 100 µL. While we mentioned that this increase might enhance the antibacterial activity against the tested pathogens, we want to emphasize that this was a theoretical assumption To reiterate, we did not conduct any experiments involving the escalation of PFFE-AgNPs concentration. The statement was included to suggest a possibility rather than to present concrete findings.

Hence the following sentence (Further increase in the concentration of PFFE-AgNPs from 25 µL to 100 µL, might increases the antibacterial activity against all the tested pathogens). has changed to  “If the concentration of PFFE-AgNPs were to be increased, it might lead to a potential increase in the antibacterial activity against all the tested pathogens”.

Point 4: In line 374, the authors mention that AgNP is effective. What do the authors refer to with this term, and what are they comparing it against to determine its effectiveness?

Response 4: In Line 374, the term "effective" is used to describe the antibacterial activity of AgNO3, albeit at a level significantly lower than that of streptomycin and the notable impact demonstrated by PFFE-AgNPs. While AgNO3 does exhibit a degree of antibacterial activity, it is important to highlight that its efficacy is notably diminished compared to the robust effects observed with streptomycin and the impressive performance of PFFE-AgNPs. This contrast underscores the varying levels of antibacterial potential among the tested substances.

Point 5: Line 377 mentions that PFFE is not effective when evaluated individually, while some lines above mention that it does have an effect.

Response 5: PFFE demonstrates a moderate antibacterial activity, as evidenced by zone of Inhibition values that are generally lower in comparison to the other tested substances.

Point 6: In line 380, the authors mention the term "potency," which is misused in the context they describe. To compare potency, a specific value of IC is used, for instance, IC50 is the one commonly employed.

Response 6: We deleted the term potency and we have included the following sentence in the revised manuscript” In the current study, streptomycin demonstrates robust antibacterial effects against all tested bacteria, surpassing all other substances examined”.

Point 7: Line 389 mentions the phrase "was explained." By whom? By the authors of this paper? Or in previous works?

Response 7: In our present experiment, we have detailed the plausible mechanism responsible for the antibacterial activity of PFFE-AgNPs. This elucidation provides insights into the underlying processes contributing to the observed antibacterial effects within the context of our study.

Point 8: Table 3 displays the effects of the inhibition zone, as they are quantitative data, the authors should present, in addition to the average, a measure of data dispersion.

Response 8: In our current study, we conducted a single round of antibacterial activity testing using five different substances against four distinct bacterial strains. The outcomes revealed notable antibacterial effects through the formation of inhibition zones. We have carried out this experiment only once due to the substantial time, labor, and meticulous efforts involved. Our primary objective was not centered around detecting subtle statistical variances. Rather, we were driven by the overarching goal of showcasing the potent antibacterial potential inherent in AgNPs.

Point 9: A significant portion of the text from lines 403 to 478 is not referenced.

Response 9: We have included the following references in the revised manuscript.

[96] Hunter, B.T., 2009. Infectious Connections. Basic Health Publications, Inc..

[97] Huemer, R.P. and Challem, J., 1997. The Natural Health Guide to Beating Supergerms. Simon and Schuster.

[98] Pham, V.H., Nguyen, Q. and Luu, V.T., 2015. Resistance (ISAAR 2015). International Journal of Antimicrobial Agents, 45(S2), pp.S59-S143.

[99] Burdușel, A.C., Gherasim, O., Grumezescu, A.M., Mogoantă, L., Ficai, A. and Andronescu, E., 2018. Biomedical applications of silver nanoparticles: an up-to-date overview. Nanomaterials, 8(9), p.681.

[100] Paladini, F., Pollini, M., Sannino, A. and Ambrosio, L., 2015. Metal-based antibacterial substrates for biomedical applications. Biomacromolecules, 16(7), pp.1873-1885.

[101] Boateng, J. and Catanzano, O., 2020. Silver and silver nanoparticle‐based antimicrobial dressings. Therapeutic dressings and wound healing applications, pp.157-184.

[102] Gurunathan, S., 2015. Biologically synthesized silver nanoparticles enhances antibiotic activity against Gram-negative bacteria. Journal of Industrial and Engineering Chemistry, 29, pp.217-226.

[103] Ahmad, F., Ashraf, N., Ashraf, T., Zhou, R.B. and Yin, D.C., 2019. Biological synthesis of metallic nanoparticles (MNPs) by plants and microbes: their cellular uptake, biocompatibility, and biomedical applications. Applied microbiology and biotechnology, 103, pp.2913-2935.

[104] Khorrami, S., Zarrabi, A., Khaleghi, M., Danaei, M. and Mozafari, M.R., 2018. Selective cytotoxicity of green synthesized silver nanoparticles against the MCF-7 tumor cell line and their enhanced antioxidant and antimicrobial properties. International journal of nanomedicine, pp.8013-8024.

[105] Engin, A.B. and Engin, A., 2019. Nanoantibiotics: A novel rational approach to antibiotic resistant infections. Current Drug Metabolism, 20(9), pp.720-741.

[106] Singh, P., Garg, A., Pandit, S., Mokkapati, V.R.S.S. and Mijakovic, I., 2018. Antimicrobial effects of biogenic nanoparticles. Nanomaterials, 8(12), p.1009.

[107] Mussin, J., Robles-Botero, V., Casañas-Pimentel, R., Rojas, F., Angiolella, L., San Martin-Martinez, E. and Giusiano, G., 2021. Antimicrobial and cytotoxic activity of green synthesis silver nanoparticles targeting skin and soft tissue infectious agents. Scientific reports, 11(1), p.14566.

Further total number of reference in this manuscript were increased from 105 to 117

Point 10: In line 508, the authors mention a lower IC50 value compared to whom? They also do not present the data that would justify observing this IC50 value.

Response 10: The IC50 stands as a representation of the concentration required to inhibit 50% of free radicals. In the context of this experiment, a lower IC50 value signifies a reduced concentration necessary to attain 50% inhibition. A diminished IC50 value in comparison to PFFE (plant extract) used alone points towards heightened antioxidant activity. This observation strongly implies that PFFE-AgNPs adeptly engage in the scavenging of DPPH radicals, showcasing an enhanced ability to counter oxidative stress. Such findings could potentially have significant implications in developing advanced antioxidant-based interventions.

Point 11: In Figure 8, it was requested to include the PFFE-AgNPS value, to which they referred as a typographical error. However, they either eliminated or did not conduct the evaluation of AgNPs alone, which is an important value that should be included. When observing effects on other variables, it is also important to include the values of the employed vehicle.

Response 11: Answer: In the present study, the silver nanoparticles (AgNPs) were synthesized using the Perilla frutescens flavonoid extract (PFFE), leading to their designation as PFFE-AgNPs. It's worth noting that in our experiment, PFFE-AgNPs encompass the entirety of our silver nanoparticle evaluation. Therefore, the distinction between AgNPs and PFFE-AgNPs is not applicable in this context. This unified terminology reflects the specific approach taken in our study, wherein the synthesis of AgNPs involves the utilization of PFFE. Consequently, the inclusion of separate AgNPs values or a distinct vehicle value doesn't arise, as PFFE-AgNPs encompass the entirety of our nanoparticle investigation.

Point 12: The authors indicate that they obtained IC50 values for the administration of PFFE in cytotoxic effects against COLO205. However, they do not present those data or graphs that would allow visualization of the evaluated concentrations and the placement of the IC50.

Response 12 : In the revised manuscript, we have provided IC50 values of PFFE (Plant extract alone) in line 551 and 552. These values reveal that PFFE alone does not exhibit significant efficacy in terms of anticancer activity against COLO205 and B16F10. Consequently, we did not pursue further investigation with PFFE alone. While we acknowledge the importance of data visualization, the extensive range of concentrations—ranging from 10 μg/mL to 500 μg/mL in increments—makes it infeasible to present every data point in graphs. Notably, the increase in concentrations did not yield substantial improvements in anticancer activity. Our primary focus remains on highlighting the remarkable anticancer potential of PFFE-AgNPs. Therefore, we omitted the presentation of PFFE-alone anticancer activity as it does not align with our objective of showcasing the effectiveness of PFFE-AgNPs in terms of anticancer activity.

Point 13: The authors indicate that the figures were standardized in terms of typography, but differences are still noticeable. For instance, Figure 2 and Figure 6 have different fonts for the x and y axes. Moreover, there are graphs where numbers appear to be cut off at the top of the y-axis.

Response 13 : Figure 2 presents the FTIR spectrum of PFFE-AgNPs, while Figure 6 provides insights into the Zeta potential measurement of PFFE-AgNPs. These figures were generated directly by their corresponding instruments, namely FTIR and DLS. It is imperative to emphasize that the originality and authenticity of these figures remain unaltered by refraining from any modifications to their letter fonts or sizes. These figures stand as accurate representations of the measurements obtained through the respective instruments, preserving the integrity of the data and ensuring the validity of our findings.

Reviewer 4 Report

I am pleased to accept your manuscript.

Author Response

Comments and Suggestions for Authors

I am pleased to accept your manuscript.

Response : Thank you for the acceptance. We are immensely pleased and express our heartfelt gratitude for the opportunity to have our paper considered in this esteemed journal.